# Inhibitors of eIF1A-ribosome interaction unveil uORF-dependent regulation of translation initiation and antitumor and antiviral effects

Daniel Hayat[1], Ariel Ogran[1], Shaked Ashkenazi[1], Alexander Plotnikov[2], Roni Oren[3], Mirie Zerbib[3], Amir Ben-Shmuel[4] & Rivka Dikstein [ID][1]✉

## Abstract

During translation initiation, eIF1A binds the ribosome through its N- and C-terminal tails, but the functional importance of this temporal interaction in mammalian cells is lacking. Using a high-throughput drug screen targeting eIF1A-RPS10 interaction, we identified inhibitors (1Ais) for eIF1A, RPS10, or both. Applying 1Ais in biochemical assays along specific and global translation experiments, we confirmed known functions of eIF1A and uncovered new roles for both eIF1A and RPS10. Specifically, the eIF1A N-terminal tail (NTT) binding inhibitors revealed the requirement of eIF1A for translation re-initiation. Moreover, a cytosine at position +5 relative to the start codon AUG, located near eIF1A-NTT in the 48S structure, enhances sensitivity to 1Ais, suggesting that the initiating ribosome recognizes a broader AUG context than the conventional Kozak. Additionally, eIF1A-specific 1Ais predominately affect cancer-related pathways. In xenograft models of ovarian cancer, these 1Ais reduced tumor growth without apparent toxicity. Furthermore, inhibition of RPS10, but not eIF1A, modulates a context-dependent regulatory translation initiation at CUG codon of SARS-CoV-2 and impedes infection. Our study underscores 1Ais as effective means to study the role of eIF1A and RPS10 in translation and suggests their targeted inhibition as potential therapies for cancer and viral infections.

**Keywords** eIF1A; Rps10; Translation Reinitiation; Ovarian Cancer; SARS-CoV-2

**Subject Categories** Pharmacology & Drug Discovery; RNA Biology; Translation & Protein Quality

## Introduction

mRNA translation is a fundamental and intricately regulated stage of gene expression. To initiate the translation of eukaryotic mRNA, the ribosome attaches the mRNA, scans the 5' untranslated region (UTR), and recognizes the correct initiation site (Hershey et al, 2019; Hinnebusch et al, 2016). This multi-step process is facilitated by various regulatory proteins and complexes termed eukaryotic initiation factors, or eIFs. The critical translation initiation stage is governed by major signal transduction pathways, proto-oncogenes, and tumor suppressors (Chu et al, 2016). The translation machinery is incredibly flexible, enabling cells to adapt to stress, activate survival mechanisms and develop drug resistance. Consequently, dysregulated translation is a common feature in diseases including cancer, viral infections, and neurodegenerative and brain-related disorders (Asati et al, 2016; Buffington et al, 2014; Costa et al, 2018; Gonatopoulos-Pournatzis et al, 2020; Green et al, 2016; Guerrero-Zotano et al, 2016; Hinnebusch et al, 2016; Kapur et al, 2017; Marquard and Jucker, 2020; Xu et al, 2020).

Most of the mRNAs initiate translation through the recognition of the 5'end m7G cap structure by eIF4F, a complex consisting of eIF4E, the cap-binding protein; eIF4G1, a large scaffolding protein that interacts with eIF4E and recruits the 43S; and the helicase eIF4A which unwinds cap-proximal secondary structures. The recruited ribosome then enters the scanning phase which is promoted by eIF1 and eIF1A (Hinnebusch, 2014), two distinct eIFs that bind the 40S subunit near the P and A sites, respectively, and promote an open 40S conformation that is scanning competent (Hinnebusch, 2017). In eIF1A the facilitation of the open conformation is mediated by its C-terminal tail, CTT (Mitchell and Lorsch, 2008). Base pairing of Met-tRNAi with an AUG triplet promotes the dissociation of eIF1 from the 40S subunit, rearrangement to a closed conformation, and scanning arrest. The 'closed' arrested conformation of the 40S is facilitated by the N-terminal tail (NTT) of the eIF1A (Mitchell and Lorsch, 2008). Thus, the two tails of eIF1A play opposing roles in scanning and AUG selection. The interaction of eIF1A with 43S is partly mediated by Rps3 and Rps10, ribosomal proteins located at the A site (Haimov et al, 2017; Sehrawat et al, 2019). How these interactions contribute to the initiation process is presently unknown.

Several studies identified gain-of-function mutations of eIF1A, particularly in the N-terminal tail, frequently found in thyroid,

[1]Department of Biomolecular Sciences, The Weizmann Institute of Science, Rehovot 76100, Israel. [2]The Nancy and Stephen Grand Israel National Center for Personalized Medicine, The Weizmann Institute of Science, Rehovot 76100, Israel. [3]Department of Veterinary Resources, Weizmann Institute of Science, Rehovot 76100, Israel. [4]Department of Infectious Diseases, Israel Institute for Biological Research, Ness-Ziona 7410001, Israel. ✉E-mail: rivka.dikstein@weizmann.ac.il

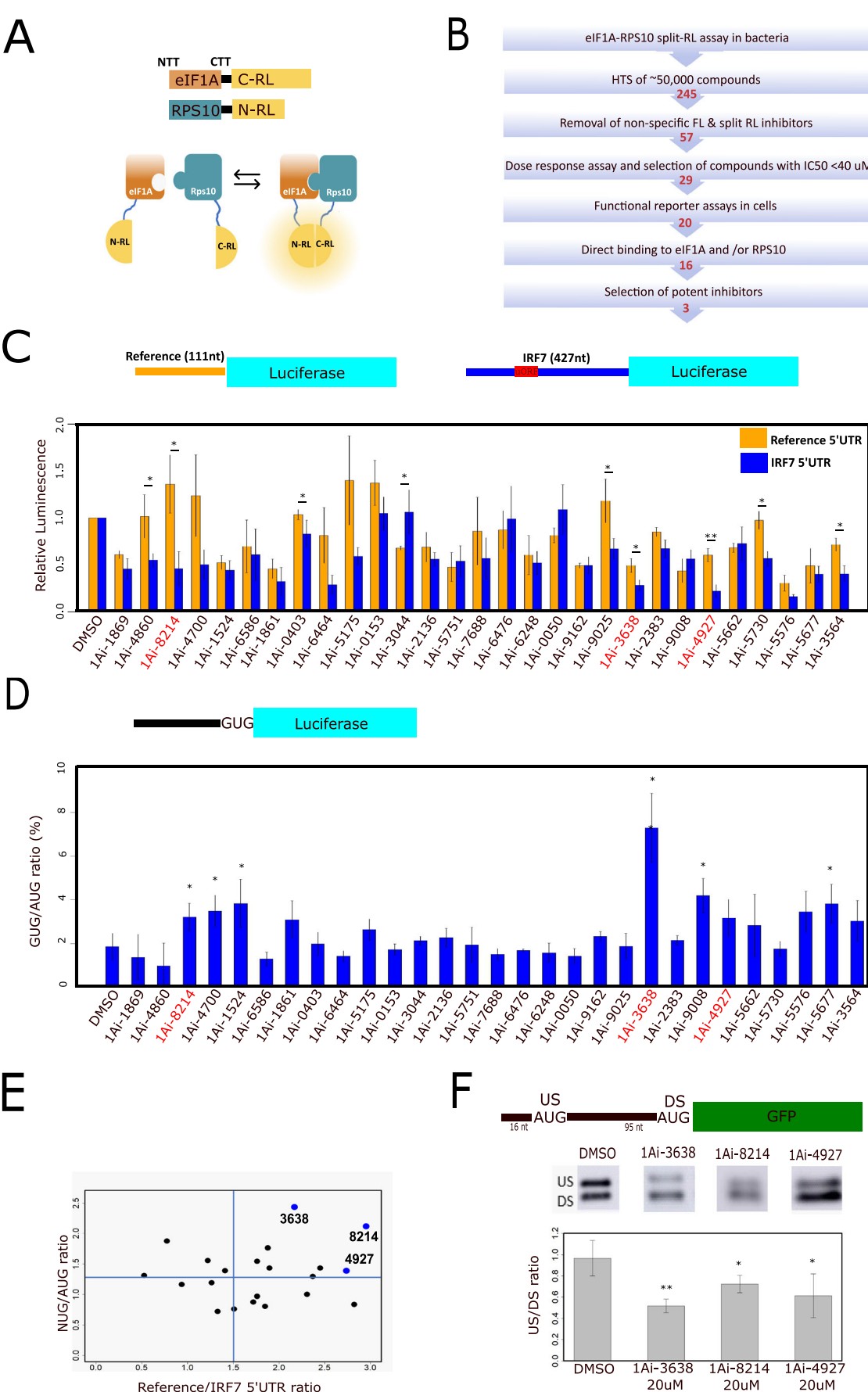

**Figure 1.   Discovery of eIF1A-RPS10 inhibitors (1Ais) using a high-throughput drug screen (HTS) in-vitro.**

(A) A scheme describing eIF1A and RPS10 split-Renilla constructs used for bacterial expression and in vitro HTS. (B) A flow chart describing the various steps of the screening process for eIF1A-RPS10 interaction inhibitors. (C) The effect of 1Ais on scanning. HEK293T cells were transfected with the described constructs that differ in their 5'UTR (the 111 nt was used as a reference, and the 427 nt is the eIF1A-sensitive IRF7) and exposed to 1Ais. Transfection efficiencies were normalized by a co-transfected GFP reporter gene. Levels of expression from reference and IRF7 5'UTR normalized to GFP were measured after overnight incubation. All of the results are relative to DMSO. Error bars represent SEM, $n \geq 4$ independent biological replicates, Student $t$ test comparing to DMSO control was performed, *$P < 0.05$. (D) The effect of 1Ais on translation initiation fidelity. HEK293T cells were transfected with the constructs that differ in their starting codon. The luciferase protein in these constructs is exclusively initiated from these starting codons. 6 h after transfection, cells were exposed to 1Ais, and the luminescence was measured after overnight incubation. Bars represent the ratio between luminescence from the AUG starting codon and GUG after normalization with co-transfected GFP. Error bars represent SEM, $n \geq 4$ independent biological replicates, Student $t$ test comparing to DMSO control, *$P < 0.05$. A similar experiment with the CUG/AUG ratio is shown in Appendix Fig. S1. (E) 2D plot summarizing the functional assays. The X axis represents short/long 5'UTR and the Y axis translation initiation fidelity (C, D); Appendix Fig. S1A). All data is presented relative to the control of the same experiment. (F) 1Ais affect leaky scanning emerging from very short 5'UTR. HEK293T cells were transfected with the GFP reporter gene schematically shown on the top. Six hours after transfection, cells were treated with 1Ais. Then, after overnight exposure to 1Ais, GFP expression was analyzed by WB. US/DS ratio was determined by densitometry and represents the long protein expression (upstream AUG, US) vs the short protein expression ratio (downstream AUG, DS). Error bars represent SEM, $n \geq 4$ independent biological replicates, *$P < 0.05$, **$P < 0.01$. Quantification of similar analysis for other 1Ais in shown in Appendix Fig. S1B. Source data are available online for this figure.

ovarian, and uveal melanoma cancers (Etemadmoghadam et al, 2017; Ewens et al, 2014; Hunter et al, 2015; Karunamurthy et al, 2016). We reported that the cancer-associated eIF1A-NTT mutants elevate the scanning-promoting activity of eIF1A, but their effect on other important functions of eIF1A, such as AUG selection and start codon fidelity is marginal (Sehrawat et al, 2019). Additionally, eIF1A is critical for cell proliferation and cell-cycle progression by promoting the translation of genes involved in cell cycle regulation (Sehrawat et al, 2019). Consequently, targeting eIF1A emerges as a promising therapeutic direction for cancer therapy.

Detailed mapping of the locations and interactions of the eukaryotic ribosome and the eIFs in the initiation complex has been achieved through structural cryo-EM and biochemical studies (Brito Querido et al, 2024; Hashem et al, 2013; Hinnebusch, 2017; Obayashi et al, 2017; Petrychenko et al, 2024; Simonetti et al, 2020; Weisser et al, 2013; Zeman et al, 2019). However, the functional significance and regulation of the dynamic protein-protein interactions during the multi-step translation initiation process, particularly in mammalian cells, remains less understood. Since mRNA translation is crucial for all cellular activities, the genes encoding most eIFs and ribosomal proteins are essential, and their genetic impairment presents a serious challenge for elucidating the in vivo importance and the specific roles of each initiation factor in mammalian cells. Moreover, certain eIFs, such as eIF4G1 and eIF1A, contain multiple independent and sometimes conflicting functional domains, making insights gained from their overall depletion less informative. An effective complementary approach for genetic perturbations involves identifying inhibitors through high-throughput screens of small-molecule libraries. This method, which we have successfully implemented and refined by targeting transcription and translation regulatory factors, utilizes recombinant proteins in a cell-free assay (Ashkenazi et al, 2016; Ashkenazi et al, 2017; Bahat et al, 2019; Sehrawat et al, 2022).

In this study, we report the identification of the first eIF1A and RPS10 inhibitors, designated 1Ais, by employing a high-throughput in vitro screen for eIF1A-RPS10 interaction. Using these newly identified compounds, we elucidated the functional significance of eIF1A and RPS10 in translation initiation. Specifically, the 1Ais that bind eIF1A disrupt the interaction of eIF1A with the ribosome in vitro and in cellular context. We demonstrated their ability to influence regulatory features mirroring those observed upon eIF1A knockdown (KD). Translatome analyses revealed that translation

initiation modulated by eIF1A-binding 1Ais is linked to a specific AUG context characterized by a C at position +5, thereby expanding the known AUG context beyond the established Kozak. Furthermore, affected genes are particularly enriched with uORFs, and further reporter gene assays established the involvement of eIF1A in re-initiation regulation, especially when scanning distance is sub-optimal. Notably, two potent 1Ais exhibit therapeutic potential by diminishing ovarian cancer development in a xenograft model, while RPS10-specific 1Ais hinders SARS-CoV-2 infection in cultured cells. This effect is highly correlated with the inhibition of a context-dependent regulatory upstream CUG initiation in the viral sub-genomic 5'UTR. Our findings uncover novel regulatory roles of eIF1A and RPS10 in translation initiation and demonstrate the implications of their inhibition on both normal and pathological conditions.

# Results

## High throughput in vitro screen for eIF1A-RPS10 interaction inhibitors

While knockdown (KD) studies of eIF1A provided valuable insights into its role in mammalian cell translation (Sehrawat et al, 2019), this approach presents several limitations. First, it fails to elucidate the functional significance of the opposing functions of eIF1A N- and C-terminal tails with respect to scanning and start codon fidelity. Second, the extended duration required for achieving efficient knockdown, 48–72 h, impairs our ability to differentiate between direct and indirect effects. To overcome these constraints, we set out to develop small-molecule inhibitors targeting eIF1A. Guided by structural and biochemical studies showing eIF1A binding to the small ribosomal subunit at the A-site and interacting directly with RPS10 through its N-terminal tail (Haimov et al, 2017; Passmore et al, 2007; Sehrawat et al, 2019), our approach focused on disrupting the interaction between eIF1A and RPS10. For this purpose, we employed the split-Renilla luciferase system, in which inactive C- and N-terminal parts of the Renilla luciferase (RL) protein are fused to eIF1A and RPS10, respectively (Fig. 1A). The interaction between eIF1A and RPS10 brings the two parts together, leading to the reconstitution of the enzymatic activity (Haimov et al, 2017). In this system, the spatial arrangement of the

two RL parts is critical for the enzymatic activity, making it sensitive not only to direct interference with the interacting proteins but also to changes in the conformation of the fused proteins. In the eIF1A-RPS10 split-RL pair, the C-terminus of eIF1A is fused to N-RL (Fig. 1A), in order to maintain the NTT accessible to RPS10 binding. This pair was cloned and expressed in bacteria as recombinant proteins and used to screen a library of ~50,000 small molecules from diverse sources (Ashkenazi et al, 2017). 245 inhibitors of the RL activity were identified, and these were analyzed against the full-length RL as well as the empty RL-N and RL-C pair, and those that affected the RL activities in these contexts were omitted. The remaining 57 molecules were then analyzed in a dose-response assay, resulting in 29 compounds (eIF1A inhibitor, 1Ai) displaying 50% inhibitory concentration (IC50) of 40 µM or less which were selected for further study (Fig. 1B; Appendix Table S1).

## 1Ais affect translation initiation regulatory features that are linked to eIF1A

To assess whether any of the 29 identified 1Ais can modulate established translation functions associated with eIF1A, particularly scanning and AUG selection, we utilized the experimental settings and reporter genes previously used to examine eIF1A KD effects on translation (Haimov et al, 2017; Sehrawat et al, 2019). For the effect on ribosomal scanning, we used two firefly luciferase reporter genes that differ in their 5'UTR length (Fig. 1C). The short 5'UTR of the reference reporter is 111 nt long and has no uORFs, and the second is derived from the IRF7 gene and is 427 nt long and bears few uORFs in its 5'UTR which may also hamper scanning. This IRF7 5'UTR was found to be downregulated by eIF1A KD (Sehrawat et al, 2019) and is expected to be more sensitive to eIF1A inhibitors. Cells were transfected with these reporters along with a GFP reporter gene used to normalize transfection efficiency. After an overnight 1Ai or DMSO (vehicle control) exposure, luciferase and GFP expression were measured, and normalized activity was determined. Eight 1Ais significantly reduced the expression of the IRF7 5'UTR reporter relative to reference 5'UTR (Fig. 1C), similar to eIF1A KD.

We next considered the importance of eIF1A in controlling start codon fidelity in yeast and mammalian cells (Martin-Marcos et al, 2011; Martin-Marcos et al, 2017; Sehrawat et al, 2019; She et al, 2023) and tested how inhibition of eIF1A by 1Ais affects the initiation from near-cognate AUG. The effect of 1Ais was analyzed using firefly luciferase reporters bearing either AUG (control), CUG, or GUG as the exclusive translation initiation sites. Co-transfected GFP reporter served as an internal control for transfection efficiency. Upon determination of the CUG or GUG to AUG ratio, we found that of the 29 compounds, 8 enhanced GUG (Fig. 1D) or CUG (Appendix Fig. S1A) utilization, while none of the 1Ai decreased the GUG or CUG initiation.

We created a 2D scatter plot to visualize the results from the two assay systems conveniently and this analysis shows that multiple 1Ais selectively decreased translation from IRF7 5'UTR and increased translation from near-cognate AUG starting codons, and the most potent of them are 1Ai-3638, 1Ai-8214, and 1Ai-4927 (Fig. 1E). We confirmed that these compounds did not exhibit any apparent effect on the mRNA levels of the five reporter genes described above (Appendix Fig. S1C,D).

Translation from an AUG preceded by a very short 5'UTR (less than 30 nt) is usually inefficient and frequently bypassed, resulting in initiation from a downstream AUG, a phenomenon known as leaky scanning. Previous (Haimov et al, 2017) and our unpublished studies revealed that eIF1A modulates leaky scanning from a cap-proximal AUG. To examine the impact of 1Ais on short 5'UTR initiation, we used a GFP reporter gene containing an AUG located 16 nt downstream from the 5' end followed by a second in-frame downstream AUG (Fig. 1F; Appendix Fig. S1B). Cells were transfected with this GFP reporter gene followed by exposure to DMSO (vehicle) and 1Ais. Initiations from upstream (US) or downstream (DS) AUGs were determined by western blot (WB) with anti-GFP antibodies and the ratio between the US and DS AUG was determined. In the DMSO control, the US/DS ratio is ~1 (Fig. 1F), while the 3 selected potent compounds and six other 1Ais significantly reduced the US/DS ratio (Fig. 1F; Appendix Fig. S1B). Thus, 1Ais enhances leaky scanning from very short 5'UTR.

## Identification of 1Ais target in the eIF1A-RPS10 complex

The identified inhibitors can either bind RPS10, eIF1A, or both at their interaction surface. To determine the binding target of the 1Ais, each eIF1A and RPS10 proteins were individually expressed, purified and fluorescently labeled. Then, the labeled eIF1A and RPS10 were each incubated with increasing concentrations of 1Ais to assess potential change in fluorescence, serving as a readout of direct binding. This approach also enables the determination of the IC50 values for the binding. Under these conditions, a clear binding of eIF1A was evident with 10 compounds (Fig. 2A; Appendix Figs. S2A and S3C). Similarly, RPS10 exhibited binding with 13 compounds (Fig. 2B; Appendix Fig. S3A,C). Apart from a few outliers that were not further studied (1Ai-2383, 5576, 7688), the binding affinities found in these assays were in the low micromolar range, similar to the values found in the screen (Appendix Fig. S3C). For 1Ai-8214 and 3638, the potent functional inhibitors, we confirmed the binding affinity using Microscale Thermophoresis (MST), a well-established ligand-protein binding assay (Jerabek-Willemsen et al, 2011). The binding curves and IC-50 were similar, and no binding was detected when eIF1A was denatured (Appendix Fig. S2D). As Rps10 was insoluble in the MST buffer, we repeated the fluorescence binding assay with its binder 1Ai-4927 using native and denatured protein. Here, too, binding was detected with the native but not denatured Rps10 (Appendix Fig. S3B). Altogether, this screen led to the identification of both eIF1A and RPS10 small molecule binders, of which 1Ai-3638 binds eIF1A, 1Ai-8214 binds both eIF1A and RPS10, and 1Ai-4927 binds only RPS10.

eIF1A interacts with the ribosome via its NTT and CTT, each adopting distinct conformation. To examine whether any of the 1Ais that bind eIF1A target its NTT or CTT, we generated eIF1A proteins with truncated NTT or CTT and used these modified proteins for the fluorescence-based binding assay described above (Fig. 2A, bottom; Appendix Fig. S2B,C). We found that two 1Ais, namely 1Ai-5175 and 1Ai-8214, exhibited no binding when tested with NTT-truncated eIF1A, while the binding affinity of a few others, such as 5677, 5576, and 4700, was significantly decreased. These findings suggest that they target the NTT fully or partially. On the other hand, 1Ai-3564 binding to eIF1A was compromised when the CTT was truncated. Notably, most of the other compounds maintained their binding affinity regardless of the truncation, suggesting their binding to the central core domain of eIF1A.

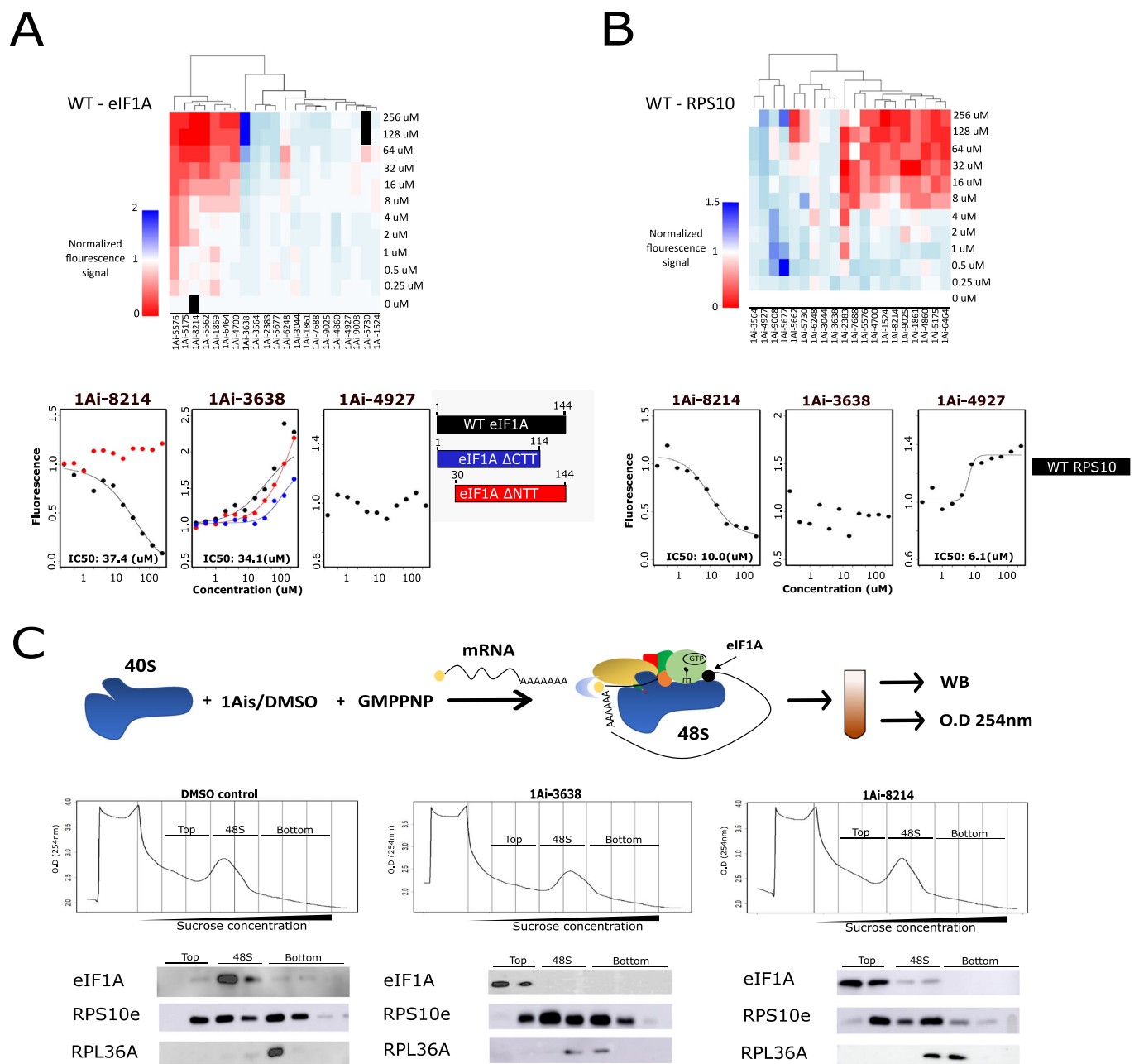

**Figure 2.   Biochemical characterization of 1Ais interaction with eIF1A and RPS10 and the 48S complex.**

(A, B) Recombinant eIF1A and Rps10 were each expressed in E. coli, purified and fluorescently labeled. Then, each of these proteins was incubated with increasing amounts of 1Ais. The binding data are presented as a heat map showing the binding of the tested 1Ais with eIF1A (**A**) and RPS10 (**B**). These data are also presented as binding curves in Appendix Figs. S2A and 3A. Binding plots of 1Ai-3638, 1Ai-8214 and 1Ai-4927 with purified and fluorescently labeled eIF1A WT or eIF1A without 30 N-terminal or C-terminal amino acids and Rps10, are shown at the bottom. All fluorescence results were normalized to 0 1Ai concentration (DMSO). The IC50 concentrations were calculated using the AAT Bioquest online tool and shown in Appendix Fig. S3C. (**C**) Rabbit Reticulocyte lysates (RRL) were pre-incubated with 1Ais, and then 48S complexes were formed by the addition of GMP-PNP and in vitro synthesized luciferase mRNA. The 48S complexes were then separated on a sucrose gradient (8–32%), and the OD at 254 nm was measured. Each gradient was separated into top, 48S, and bottom fractions according to the 48S peak location, and samples from each fraction were subjected to TCA precipitation followed by WB testing for the presence of the indicated proteins. Source data are available online for this figure.

## 1Ai-3638 and 1Ai-8214 inhibit eIF1A association with the 48S in vitro and in cells

Considering that eIF1A binding to the ribosome is central to its function, we examined how the potent 1Ais that bind eIF1A affect its

interaction with the 48S translation complex. Ribosomal complexes were formed in vitro using rabbit reticulocyte lysate (RRL) as a source for ribosomes and eIFs. After 5 min of incubation with 1Ai, ribosomal complexes were supplemented with in vitro synthesized and capped model mRNA and GMP-PNP, a GTP analog used to stall translation

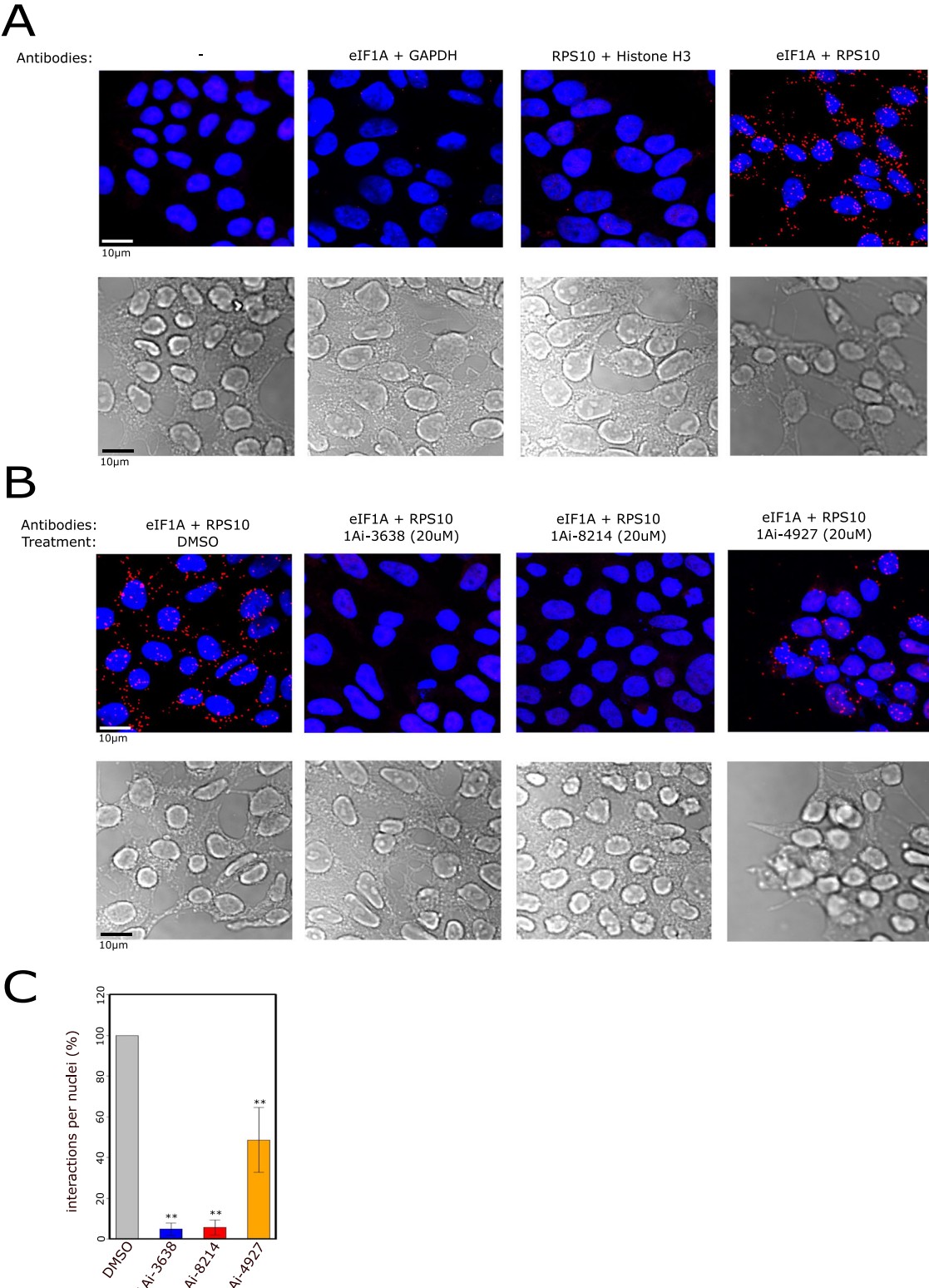

at the 48S step (Fig. 2C). The 48S was then fractionated on a sucrose gradient and identified by OD at 254 nm. Western blot with antibodies against RPS10, eIF1A and RPL36 confirmed that the peak marks the 48S while the 60S large subunit is found in heavier fractions (Fig. 2C).

In a control experiment, in which the same reaction was carried out in the presence or absence of mRNA, the 48S peak, including eIF1A and RPS10, is detected only in the presence of the mRNA (Appendix Fig. S4A). While in the DMSO-treated complex eIF1A co-migrates with

◄ **Figure 3. 1Ais disrupt eIF1A interaction with RPS10/ribosome in cells.**

(A–C) HEK293 cells were grown on a glass slide, exposed to the selected 1Ai for 3 h, fixed and subjected to Proximity ligation assay (PLA). Cells were reacted with the indicated matching antibody pairs, control pairs and eIF1A and RPS10 pair (A) and the DMSO and 1Ai-treated cells with the eIF1A and RPS10 pair (B). The samples were treated according to PLA protocol and stained with DAPI to visualize cellular orientation and numbers. PLA dots (red), nuclei (blue) were visualized using confocal microscopy. Each picture is a merge of ~15 pictures covering all cell volume. (C) Quantification of PLA results for eIF1A and RPS10 antibodies and 1Ais treatment, $n = 4$ independent biological replicates, error bars represent SEM, ** denotes $P < 0.01$ according to Student $t$ test. Source data are available online for this figure.

the 48S, upon 1Ai-3638 and 1Ai-8214 treatments, eIF1A location is shifted and is found mostly in the top fractions of the gradient, suggesting that both compounds prevent eIF1A engagement with the 48S ribosomal complex. A similar experiment with the RPS10-specific 1Ai-4927 revealed no change in eIF1A association with the 48S (Appendix Fig. S4B).

We performed initial experiments aimed to determine eIF1A association with the 40S in cells either by sucrose gradient sedimentation or co-IP. However, we could not detect this interaction suggesting that it is either not maintained upon cell lysis or it is below detection level. Therefore, we turned to another approach to examine whether 1Ais can affect the interaction between eIF1A and RPS10/ribosome in cells. We employed the proximity ligation assay (PLA) that enables the visualization and quantification of specific protein interactions within cells (Bagchi et al, 2015). In this method, specific antibodies are used to bind to the target proteins of interest. Then, DNA oligonucleotide-labeled secondary antibodies are introduced and bind to the primary antibodies. If the two proteins are in proximity, they bring the DNA oligonucleotides close together. Then, DNA ligation, rolling circle amplification, and detection steps are performed, resulting in the generation of distinctive signals that can be seen as fluorescent dots by fluorescence microscopy. HEK293T cells were treated with 1Ais for 3 h, fixed, and reacted with eIF1A and RPS10 antibodies according to PLA protocol (Bagchi et al, 2015). In addition, cell nuclei were stained with DAPI to assess cell orientation and numbers. In parallel, we determined the background noise; we used each antibody (anti-eIF1A or RPS10) with another matching control antibody. These experiments revealed a strong signal when using eIF1A and RPS10 antibodies while almost undetected fluorescence when replacing one antibody with the control antibody (Fig. 3A), confirming their close proximity in the cell. Under these conditions, 1Ai-3638 and 1Ai-8214 diminished the RPS10-eIF1A interaction signal to almost background levels, while 1Ai-4927, which binds only RPS10, reduced the interaction but to a lesser extent (Fig. 3B,C). These findings confirm that these compounds affect eIF1A interaction with the small ribosomal subunit.

## Translation initiation modulation by 1Ais is linked to AUG context and regulation by uORFs

To assess the impact of 1Ais on cellular translation, HEK293T cells were treated with DMSO, 1Ai-3638, 1Ai-8214 and 4927 for 3 h, then cell lysate was fractionated on sucrose density gradient sedimentation. The analysis of the polysome profiles of the different samples revealed an accumulation of the 80S mono-ribosome upon 1Ai-3638, 1Ai-8214, and 1Ai-4927 treatments compared to DMSO with a significant reduction in the polysome to monosome ratios (Fig. 4A). These effects are consistent with impairment of translation initiation by these 1Ais. The effect of 1Ai-4927 on polysomes is more pronounced, raising the possibility

that it also affects translation elongation, which is consistent with its ability to bind Rps10.

To gain a global quantitative insight into the impact of the eIF1A-specific 1Ai-3638 and 1Ai-8214 on translation, we treated HEK293T cells for 3 h with the compounds and then subjected them to Ribo-seq (deep sequencing of ribosomal protected RNA fragments) and total mRNA sequencing that in combination enable calculation of the changes in translation efficiency (Ingolia et al, 2012). Following 1Ais treatment, we observed a global reduction in the relative abundance of mRNA in the treated samples (Fig. 4B). Metagene profiles revealed a strong reduction in ribosomal footprints in the coding region in the 1Ais treated samples (Fig. 4C) and the anticipated 3-nucleotide (nt) periodicity (Appendix Fig. S5C), suggesting inhibition of translation that is consistent with the polysome profiles. The TE determination revealed that the 1Ai-3638 and 1Ai-8214 treatments led to ≤1.5 folds translational downregulation of 246 and 325 genes, respectively, and fewer upregulated genes, 21 and 11, respectively (Fig. 4D). Interestingly, we observed a significant overlap between the downregulated genes affected by 1Ai-3638 or 1Ai-8214 (17%, $P < 10^{-5}$) (Fig. 4E). This finding diminishes the likelihood that the downregulation of the common mRNAs can be attributed to off-target effects triggered by two chemically distinct small molecules.

Considering the importance of eIF1A for the regulation of AUG selection, we examined the sequence context of the CDS start codons in 1Ai-3638 or 1Ai-8214 translationally downregulated mRNAs (Fig. 4F,G). In comparison with the unaffected mRNAs, in both 1Ai-3638 or 1Ai-8214 downregulated mRNAs, there is an enrichment of an A at position −3 while the important +4 G position is underrepresented (Fig. 4F,G). Additionally, we found that a C at position +5 is enriched among the downregulated genes of both (Fig. 4F,G). To understand the basis for cytosine enrichment in position +5, we examined the cryo-EM structure of the mammalian 48S complex in which eIF1A NTT is partially visible (PDB: 6YAL, (Simonetti et al, 2020)). Interestingly, in this structure, the RNA binding core domain is close to position +4 via W70 (4.03 Å), while eIF1A's NTT is very close to the cytosine in position +5 with a similar proximity (3.5 Å, Fig. 4H).

To examine further the importance of the +5 C position for the response to 1Ais, we used the IRF7 reporter gene. A C was introduced to the original weak AUG context at the +5 position. In addition, the IRF7 was modified to a strong Kozak context without or with a +5 C, as shown schematically in Fig. 4I. The change in the +5 C modified the N-terminal amino acid sequence and, conse-quently, protein stability due to the N-end rule; we, therefore, examined the effect of the compounds on these constructs by polysome profiling. Each of these reporter genes was barcoded and then co-transfected into cells, and the ratio of each mRNA in the polysome to total mRNA was determined by RT-qPCR using barcode-specific primers. The results revealed that the inhibitory

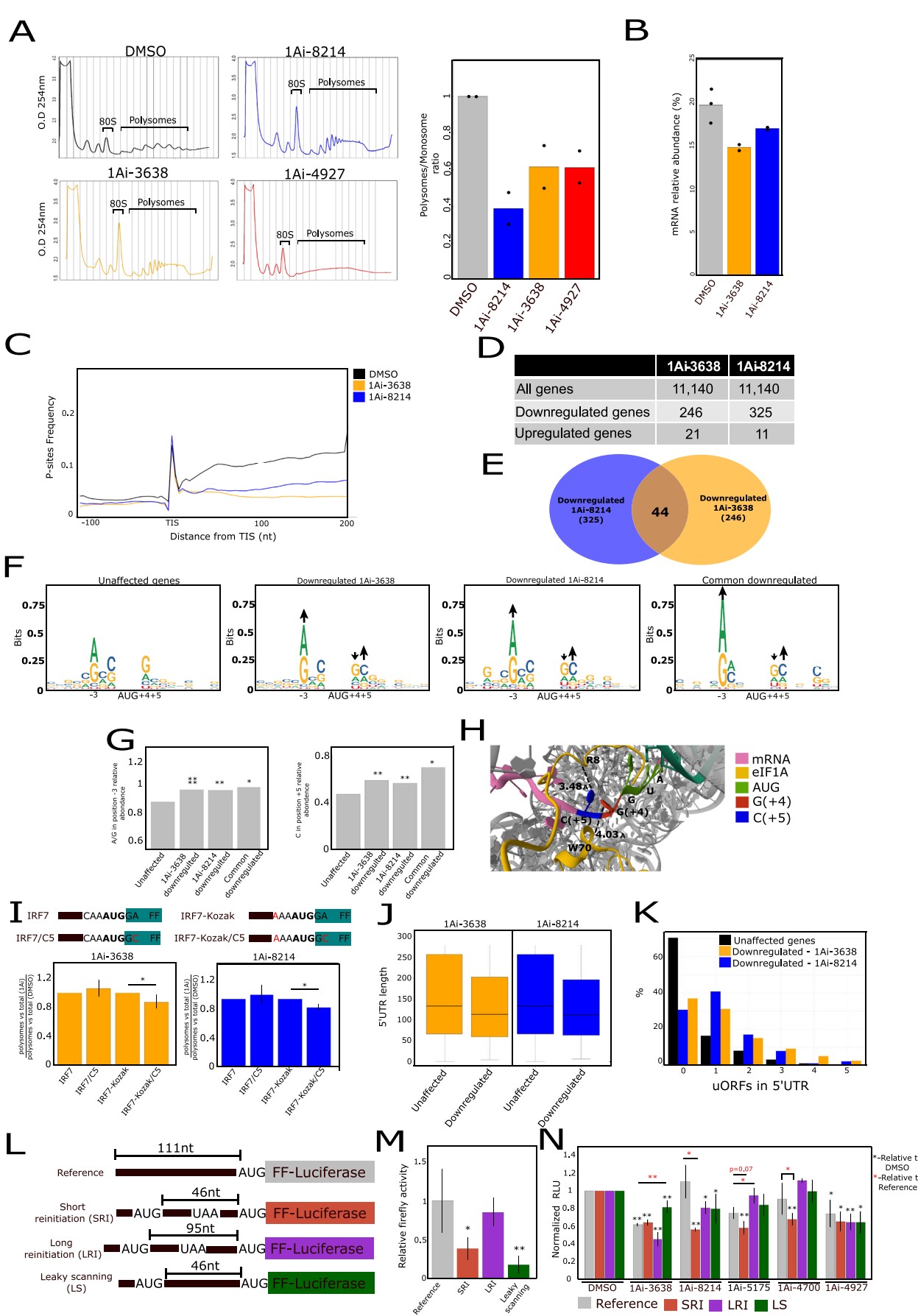

**Figure 4. 1Ais impairment of translation initiation is related to AUG context and regulation by uORFs.**

(A) HEK293 cells were treated with DMSO or 1Ais for 3 h and subjected to polysome profiling by fractionating cell lysates by centrifugation on a sucrose density gradient (10–50%) followed by OD measurement (254 nm) of each fraction. The quantified polysomes to monosomes ratio is shown on the right. The bars represent the mean, $n = 2$ independent biological replicates. (B–I) The effect of translation inhibition by 1Ais on global translation using Ribo-seq. HEK293T cells were treated with DMSO or 1Ai-3638 and 1Ai-8214 for 3 h then cycloheximide addition, followed by preparation of ribosome footprinting libraries, deep sequencing and analysis according to standard RF protocol. In parallel the same samples were also subjected to RNA-seq for total mRNA level measurements. (B) Relative mRNA abundance after 1Ais or DMSO treatment. The bars represent the mean, $n = 2$–3 independent biological replicates. (C) Metagene analysis. The p-site in each ribosome footprint was located and the ribosomal occupancy of each area on the mRNA was calculated and normalized to mRNA total counts. Data is normalized to the TIS peak. (D) A list summarizing the number of genes that are differentially translated relative to DMSO control (relative TE > 1.5 or relative TE < 0.5). (E) Venn-diagram representing the intersection between genes downregulated by 1Ai-3638 and 8214. The significance of the overlap is $P < 10^{-5}$ and was calculated by Randomization test. (F) Analysis of the nucleotide context of the annotated start codon of unaffected and 1Ai-3638 and 1Ai-8214 downregulated genes and the overlapping downregulated gene set. Black arrows indicate affected nucleotides. (G) The frequencies of the C + 5 and A/G-3 in 1Ais unaffected and downregulated mRNAs; *$P < 0.05$, *$P < 0.01$, Randomization test. (H) Cryo-EM structure of yeast eIF1A, including its NTT in the entry channel of the ribosome (PDB:6ZMW). The distances were measured by Pymol. (I) Three barcoded firefly reporter mutants were generated (scheme on the top) based on the IRF7 5'UTR (IRF7/Kozak, IRF7-C5, IRF7/Kozak-C5). These constructs were co-transfected to HEK293T cells and, 6 h after transfection, incubated with either DMSO, 1Ai-3638, or 1Ai-8214 (20 μM). Cell lysates were fractionated by sucrose gradient sedimentation. RNA was extracted from the polysomes and total fractions followed by DNase I treatment and RT-qPCR using primers specific for each barcode. Data is presented as the ratio of polysomes vs total RNA abundance after 1Ai treatment relative to DMSO; $n = 3$ independent biological replicates, *$P < 0.05$, $t$ test, error bars represent SEM. (J) 5'UTR length distribution among downregulated and unaffected genes after 1Ais treatment in which the center = median, box ends = interquartile range (IQR), bottom whisker = minimum. Unchanged are 10,569, 3638 down 246 and 8214 down 325 genes. (K) The number of uORFs per gene among downregulated and unaffected genes. The presence of uORFs was determined based on previous RF data (Sehrawat et al, 2022). (L) A scheme of firefly luciferase reporters that include the reference 5'UTR (Fig. 1C), which was altered to have an uORF starting 46nt (short reinitiation, SRI) or 95 nt (Long reinitiation, LRI) or an AUG 46nt before the main ORF without an in-frame stop codon to create a reporter with leaky scanning (LS). (M) The reference and uORF-bearing reporters were transfected to HEK293T cells together with RL, which served to normalize translation efficiency. The reporter activities are represented as firefly vs RL; $n \geq 3$ independent biological replicates, error bars represent SEM. *$P < 0.05$, **$P < 0.01$, Student $t$ test. (N) Firefly Reporters (J) were transfected to HEK293T cells together with normalizing RL. Six hours after transfection, DMSO or 1Ais at 20uM (except 1Ai-5175 at 5uM) were added for overnight incubation. Results are represented as firefly vs. RL normalized to DMSO in each biological replicate; $n = 4$, error bars represent SEM. *$P < 0.05$, **$P < 0.01$, Student $t$ test. Source data are available online for this figure.

effect of the compounds on IRF7 and IRF7-C5 is similar. However, both 1Ai-3638 and 1Ai-8214 inhibitory effects were moderately yet significantly increased with the IRF7-Kozak/C5 compared to IRF7-Kozak (Fig. 4I). These findings suggest that Kozak with a +5 C is particularly sensitive to 1Ais, reinforcing the Ribo-seq-derived AUG context.

We next analyzed other potential features of the 1Ais down-regulated mRNAs related to scanning. As the IRF7 5'UTR is sensitive to 1Ais and is both long and has uORF (Fig. 1C), we determined the 5'UTR length and the presence of uORFs in the unaffected and downregulated genes. uORFs data was derived from TIS-seq data of DMSO-treated HEK293T cells (Sehrawat et al, 2022). While the 5'UTR length of the downregulated and unaffected mRNAs is very similar (Fig. 4J), we found enrichment of uORFs among the 1Ai-3638 and 1Ai-8214 downregulated mRNAs relative to the unaffected gene set (Fig. 4K). Further analysis of the uORF initiation context revealed that while the initiation site of the unaffected mRNAs is mostly NUG, in the downregulated mRNAs, the first nucleotide is enriched with A/C (Appendix Fig. S5D).

To further elucidate eIF1A involvement in uORF regulation, we utilized the reference FL reporter gene described above (Fig. 1C) to generate three constructs. In the first and second, we inserted an out-of-frame uAUG 46 or 95 nucleotides before the main ORF, respectively, followed by a downstream stop codon generating an uORF of 10 amino acids. In these constructs, the translation from the main ORF could be either by reinitiation or leaky scanning. In the third construct, this out of frame uAUG lacks a stop codon, causing initiation from the main ORF to exclusively occur via leaky scanning (Fig. 4L). These reporters were transfected into HEK293T cells together with RL luciferase reporter for normalizing transfection efficiency. The construct bearing an uORF with a short re-initiation distance displayed decreased reporter activity while the one with the long re-initiation retained most of it (Fig. 4M). The

absence of a stop codon resulted in a more pronounced inhibitory effect, likely due to the inability to perform reinitiation (Fig. 4M). We next examined these reporters in the presence of 1Ais. Specifically, 1Ai-3638 decreased all 5'UTRs with exclusive leaky scanning being less affected (Fig. 4N), consistent with its stronger potency in inhibiting multiple aspects of translation associated with eIF1A (Fig. 1E). Interestingly, 1Ai-8214 exhibits a clear differential effect, more strongly inhibiting the activity of 5'UTR bearing uORF with short re-initiation distance (Fig. 4N). Since 1Ai-8214 binds the NTT of eIF1A, we examined the other NTT-binding 1Ais, namely 1Ai-5175 and 1Ai-4700, and found that both inhibit the short-distance reinitiation (Fig. 4N). Conversely, no differential effects were observed with the RPS10-binding 1Ai-4927 (Fig. 4N). These results suggest that eIF1A plays a role in uORF regulation through reinitiation mechanisms.

## 1Ai-3638 and 1Ai-8214 decrease the development of ovarian cancer in a xenograft model

To identify the major biological themes affected by 1Ai-3638 and 1Ai-8214, we analyzed the translationally downregulated genes using Gene Analytics (Ben-Ari Fuchs et al, 2016). We found that cell cycle progression and various cancer pathways are significantly enriched among the affected genes of both compounds (Fig. 5A). Considering the prevalence of eIF1A gain of function mutations in certain cancer cell types, including uveal melanoma and ovarian carcinoma (Etemadmoghadam et al, 2017; Ewens et al, 2014; Karunamurthy et al, 2016), and the essentiality of eIF1A for cell proliferation (Sehrawat et al, 2019), we first assessed the ability of 1Ais to influence the viability and proliferation of an ovarian cancer cell line, OVCAR8. Cells were exposed to increasing concentrations of 1Ai-3638 and 1Ai-8214 for 72 h with 1Ais refreshment every 24 h. This treatment led to reduced OVCAR8 cell viability by both 1Ai-3638 and 1Ai-8214 (Fig. 5B) as also seen for other proliferating

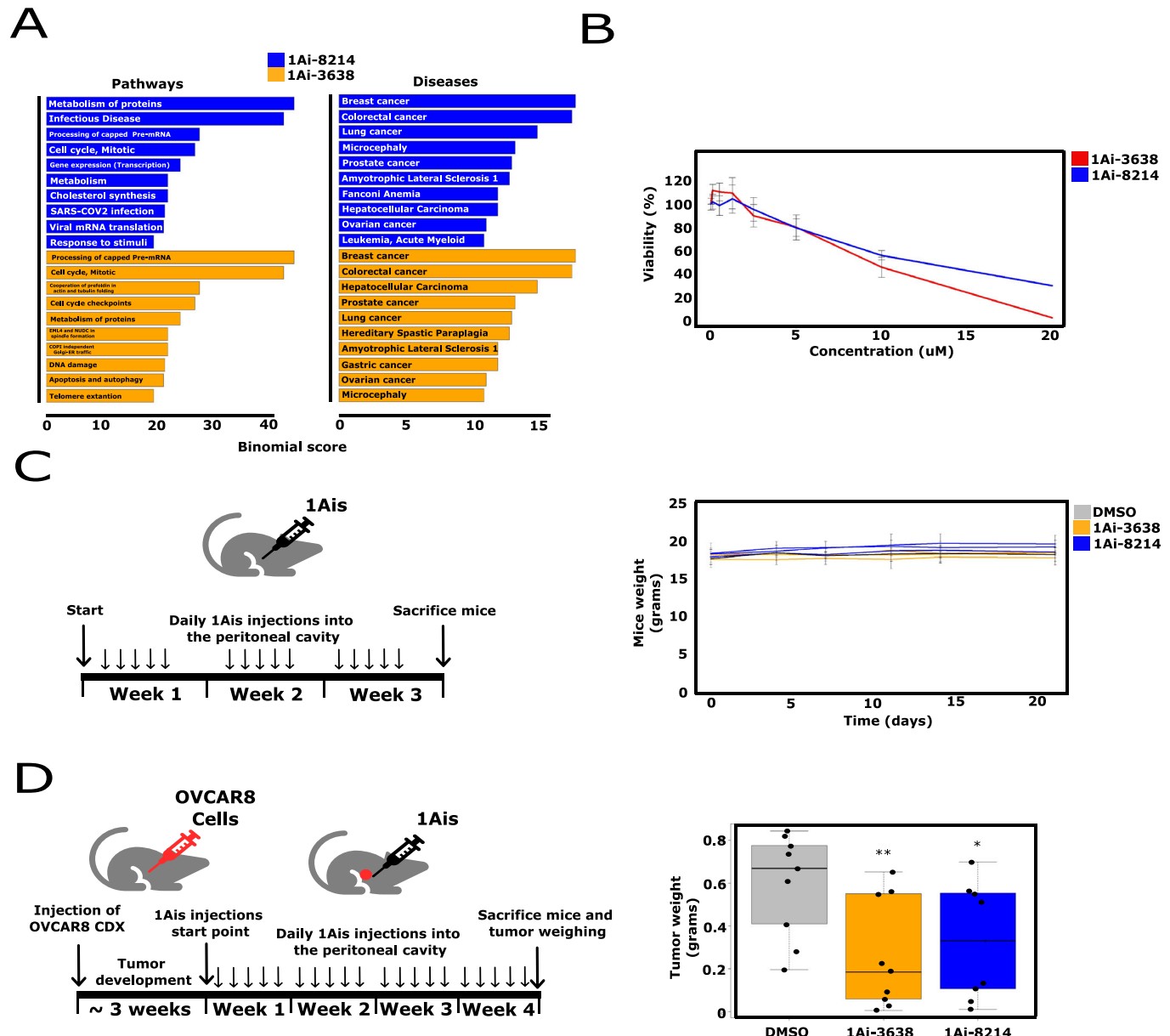

**Figure 5. 1Ai-3638 and 1Ai-8214 decrease ovarian tumor growth in xenograft mouse model.**

(A) Gene and pathways enrichment analysis of 1Ai-3638 and 1Ai-8214 downregulated genes by the Gene Analytics tool. (B) OVCAR8 ovarian carcinoma cell line was incubated with increasing concentrations of 1Ais for 72 h, and then cell viability was assessed. Viability (100%) is normalized to 0 concentration (DMSO), $n = 3$ independent biological replicates, and error bars represent SD. (C) 1Ais toxicity assessment. Mice weight was measured at different time points after 1Ai-3638 and 1Ai-8214 subcutaneous injection for 3 weeks (assessed every 3–4 days, 3 mice per concentration, 21 mice in total) (D) OVCAR8 cell-derived xenografts weight after 1Ais treatment (10 mice per group). OVCAR8 cells were injected into the ovaries of immunocompromised female mice. Out of ten mice in each group (1Ai-3638,1Ai-8214 and DMSO) nine were included as they presented an ovarian tumor. 1Ais were injected into the mice's peritoneal cavity five times a week for 4 weeks. Finally, mice were sacrificed, and tumor weight was measured. Results are presented as boxplots in which the center = median, box ends = interquartile range (IQR), bottom whisker = minimum and upper whisker= maximum, $n = 9$. Weight monitoring of mice during the experiment is presented in Appendix Fig. S3. *$P < 0.05$, **$P < 0.01$, Student $t$ test. Source data are available online for this figure.

cells. Sequentially, we asked if 1Ais decrease OVCAR8 viability in an animal context using xenografts. Considering the importance of eIF1A for translation in general, we first tested the toxicity of different concentrations of 1Ais in mice. C57 black mice were injected with DMSO control or 1Ai-3638 and 1Ai-8214 at 3 different concentrations (3 mice per concentration) up to 8 mg/kg

five times a week for 3 weeks. No significant effect on the mice's well-being or any loss of weight was observed (Fig. 5C), indicating no apparent toxicity. This finding prompted us to examine whether 1Ai-3638 and 1Ai-8214, at the highest concentration, can impact tumor growth of an OVCAR8 cell-derived xenograft (CDX) model. Thirty immunocompromised female mice were injected with 5

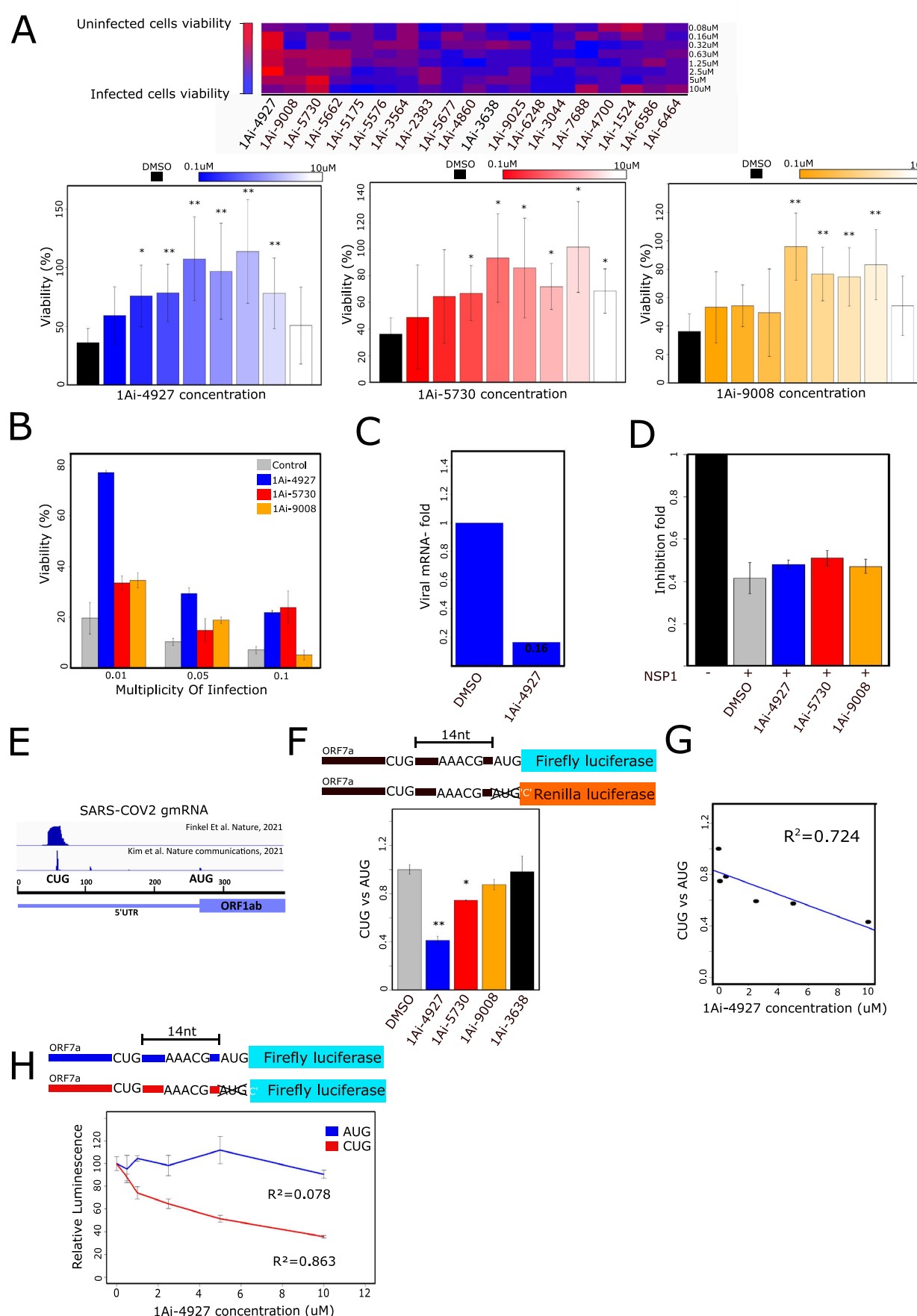

Figure 6.   1Ai-4927 decreases SARS-COV-2 infection by selective inhibition of a context-specific regulatory CUG element.

(A) 1Ais enhances cell viability after SARS-COV-2 infection. Vero6 cells were incubated with increasing concentrations of 1Ais and infected by the SARS-COV-2 virus. After 72 h, cell viability was measured by counting live cells. The 1Ais that presented a significant effect are presented in Bar plots. 100% represent uninfected cells and the black line cells infected with SARS-COV2 but treated with DMSO. Error bars represent SD of the mean, *$P < 0.05$, **$P < 0.01$, $n = 3$ independent biological replicates. All 1Ai viability plots are shown in Appendix Fig. S4A. (B) Vero6 cells infected by SARS-COV-2 at different multiplicities of infection and treated by 1Ais as described in Fig. 6A. Error bars represent the mean ± SD, $n = 2$ independent biological replicates. (C) mRNA sample was taken from Vero6 cells rescued by 1Ai-4927 (A) and tested by q-PCR for viral mRNA quantity. (D) The effect of 1Ais on NSP1-mediated inhibition of translation. HEK293T Cells were transfected with NSP1 and a GFP reporter. Cells were incubated overnight with 1Ais, $n = 6$ for DMSO and no NSP1 and $n = 2$ for 1Ais treatments. Data for all tested 1Ais is presented in Appendix Fig. S4B. (E) Presentation of ribosomal footprints from SARS-COV-2 infected cells and mapped to the genomic 5'UTR at the regulatory CUG derived from (Finkel et al, 2021; Kim et al, 2021). (F) SARS-COV-2 ORF7a 5'UTR was inserted upstream to a firefly and Renilla luciferase reporters. The Renilla reporter ORF was aligned with the regulatory CUG 14 by adding one nucleotide after the main AUG. HEK293T cells were co-transfected with both reporters and, after 6 h, incubated with DMSO or 1Ai-4927 overnight. The ratio between the Renilla and Firefly reporters was measured and normalized to DMSO control. Left panel, error bars represent SEM, **$P < 0.01$ $n = 3$ independent biological replicates. (G) The same reporters as Fig. 6F were incubated with increasing concentrations of 1Ai-4927, $n = 3$. (H) Firefly luciferase is driven by the ORF7a 5'UTR with the CUG as the exclusive initiation was constructed. HEK293T cells were transfected with either AUG or CUG starting codon firefly reporter, and 6 h after transfection, cells were incubated with increasing concentrations of 1Ai-4927 overnight. The luciferase signal was measured for CUG and AUG separately and normalized to DMSO. Error bars represent SEM, $n = 4$ independent biological replicates. Source data are available online for this figure.

million OVCAR8 cells directly into the ovary. When tumors were formed after about 3 weeks (9 out of 10 mice in each group), a treatment of 8 mg/kg of either 1Ai-3638 or 1Ai-8214 or DMSO (control) was given by injection to the peritoneal cavity five times a week. After 4 weeks of injections, mice were sacrificed, and tumors were removed from the ovaries and weighed. In both 1Ais treatment groups, a clear reduction in tumor size and weight was observed (Fig. 5D). The mice did not seem to suffer from any treatment-specific side effects (Appendix Fig. S6). These findings suggest that targeting eIF1A by small molecules has the potential to reduce ovarian tumor growth.

## RPS10-binding 1Ais reduce SARS-CoV-2 propagation and impede regulatory CUG initiation

The translation of SARS-CoV-2-encoded mRNAs by host ribosomes is crucial for virus replication. While the genomic 5'UTR that controls early viral translation acts in a cap-independent manner, the sub-genomic 5'UTR of late genes translates via the canonical mode of translation and is sensitive to eIFs perturbations (Slobodin, 2022). To examine the potential of 1Ais as antiviral agents, we treated Vero E6 cells with increasing concentrations of the 19 biologically active 1Ais, infected them with SARS-CoV-2, and monitored the cytotoxic effect of the virus after 72 h. We found that many of the 19 tested 1Ais enhanced the survival of the infected cells to variable degrees, while at the highest concentrations, they were toxic for cells (Fig. 6A; Appendix Fig. S7A). Three 1Ais, including 1Ai-4927, 5730, and 9008, rescued cell survival following SARS-CoV-2 infection to the level of uninfected cells (Fig. 6A). Interestingly, these 1Ais bind exclusively to RPS10 (Fig. 2B). These findings suggest that SARS-CoV-2 is substantially more sensitive to RPS10 inhibition than the host cells. We then tested if these three compounds could rescue cell viability at a variable multiplicity of infection (MOI) of SARS-CoV-2. Both 1Ai-4927 and 1Ai-5730 increase live cell count at all tested MOIs, but 1Ai-4927 is more potent (Fig. 6B). We confirm this finding by analyzing RNA samples from the infected cells after 1Ai-4927 exposure by qPCR for viral RNA presence, which was indeed reduced to 16% relative to the cells treated with DMSO control (Fig. 6C).

To understand the underlying basis of the 1Ai-4927 effect, we first focused on NSP1, the first protein encoded by SARS-CoV-2 and acts to inhibit translation of the host but not the viral mRNAs (Burke et al, 2021; Finkel et al, 2021; Slobodin et al, 2022; Thoms

et al, 2020). We examined whether 1Ais can relieve translational inhibition induced by NSP1. Cells were co-transfected with the NSP1-sensitive GFP reporter gene together with NSP1 and then treated with 1Ais. None of the 1Ais, including 1Ai-4927, could affect the NSP1 inhibitory activity (Fig. 6D; Appendix Fig. S7B).

To further investigate the mechanism of SARS-CoV-2 repression by 1Ais, we considered a translation regulatory element present in the sub-genomic 5'UTR consisting of a near-cognate CUG codon (Fig. 6E) and a motif (AAACG) located 10–14 nucleotides downstream (Havkin-Solomon et al, 2023). This element is located close and upstream to the AUG start codon of the sub-genomic mRNAs and was shown to be regulated by the mRNA binding activity of RPS3 at the entry channel (Havkin-Solomon et al, 2023). Considering that 1Ai-4927 targets RPS10 which is also part of the entry channel and interacts directly with RPS3 (Haimov et al, 2017), we asked whether 1Ais can modulate the translation from this regulatory element. To this end, we designed 2 reporter genes based on the 5'UTR of SARS-CoV-2 ORF7a. In one, the AUG is in frame with the firefly luciferase but out of frame relative to the CUG, while in the second, the Renilla luciferase is in frame with the CUG codon, and the AUG is out of frame (Fig. 6F). The choice of the ORF7a is because it is the highest expressing ORF in which the CUG and AUG are not in frame (Kim et al, 2021). HEK293T cells were co-transfected with the two reporters and exposed to the 3 RPS10-selective 1Ais or to 1Ai-3638, which binds exclusively to eIF1A. Then, luminescence was measured, and the ratio of firefly (AUG) to Renilla (CUG) luciferases was determined. Unexpectedly, we found that the RPS10-specific 1Ais (1Ai-4927, 5730 and 9008) decreased the CUG/AUG ratio to variable degrees while 1Ai-3638, the most potent eIF1A inhibitor, had no effect (Fig. 6F). Further examination of the most effective compound, 1Ai-4927, confirms a dose-response inhibition of the CUG/AUG ratio (Fig. 6G). To address which of the two initiation sites are targeted by 1Ais, namely whether AUG is increased or CUG is decreased, and to exclude a potential translation elongation effect that may be exerted by the two different reporters, we also generated a firefly luciferase reporter with the CUG as the initiation codon. Each of the AUG and CUG firefly reporters was transfected and then treated with increasing concentrations of 1Ai-4927 for 24 h. The results revealed that the 1Ai-4927 primarily decreased translation from the regulatory CUG, while translation starting from the AUG was

not significantly affected (Fig. 6H). The inhibitory effect of CUG contrasts with the findings in Fig. 1D, where 1Ais, including 1Ai-4927 and 1Ai-3638, generally enhanced CUG initiation and reduced initiation fidelity.

To further investigate the underlying basis of the 1Ai-4927 effect on this regulatory element, we generated three additional constructs. In the first, the CUG initiation was modified to CUA (Appendix Fig. S8A). This resulted in an activity comparable to background levels (Appendix Fig. 8B). In the second, the CUG was modified to AUG, which substantially enhanced the activity as expected (Appendix Fig. S8B). In the third, the AAACG motif (RPS3RS) was mutated to TTTCG, which also diminished activity almost to the background (Appendix Fig. S8B), confirming its importance. Therefore, it was possible to examine the effect of the 1Ai only on the AUG mutant. The results revealed that the extent of inhibition by 1Ai-4927 was significantly reduced compared to the original CUG (Appendix Fig. S8C). These findings confirm the importance of both CUG and AAACG sequences for the activity and the response to RPS10-specific 1Ais.

# Discussion

In this study, we combined a high-throughput in vitro screen targeting eIF1A-RPS10 interaction with functional assays and identified the first eIF1A and RPS10 inhibitors (1Ais). These compounds provided valuable insights into the unknown roles of these proteins in the regulation of translation initiation and the implications of their inhibition on cellular processes and pathological states. By classifying the 1Ais into three major classes according to their binding target, we were able to dissect and uncouple the diverse activities of these proteins. The development of small-molecule inhibitors targeting different domains of eIF1A and RPS10 not only complements alternative methods, such as knockdown (KD) experiments, but also addresses their inherent limitations. For example, 1Ais enable fast and effective target inhibition, overcoming the prolonged duration needed for an efficient knockdown and resolving the challenge of distinguishing between direct and indirect effects associated with KD studies. Furthermore, we demonstrated that the 1Ais can precisely target a specific domain and function within the eIF1A-RPS10 complex. Thus, the 1Ais represent a powerful new tool for elucidating the intricate roles of these proteins in cellular processes.

The identification of a large number of potentially specific inhibitors can be explained by the relatively weak and transient nature of the eIF1A-RPS10 interaction during translation, making it more susceptible to displacement. The majority of the compounds exhibited direct binding to eIF1A, RPS10, or both, and using 3 distinct reporter systems, we found that many of these compounds affect translation in a manner dependent on start codon identity and AUG proximity to the 5'end, akin to the effects observed upon eIF1A KD. Focusing on the most potent inhibitors with respect to their effect on scanning, start codon fidelity, and direct binding, we show that 1Ai-3638 binds the eIF1A core domain, 1Ai-8214 targets both eIF1A NTT and RPS10, and 1Ai-4927 binds exclusively to RPS10. These compounds diminish the interaction of eIF1A with the small ribosomal subunit in cells, but whether they affect the 43S, the 48S, or both cannot be inferred from these assays. Moreover, we cannot exclude the possibility that eIF1A

can still bind the translation initiation machinery with a lower affinity via other components. As evident from polysome profiling and Ribo-seq analyses, these compounds inhibit translation initiation. Translation efficiency calculations demonstrated the downregulation of specific genes by 1Ai-3638 and 1Ai-8214, with a significant overlap between affected mRNAs, indicative of a shared target within the translation machinery. eIF1A has several functional domains, two with opposing functions (NTT and CTT). Therefore, it is expected that targeting different domains may result in somewhat differential effects on various mRNAs. Consistent with this, although there is a significant number of overlapping genes, most of the affected genes are distinct, revealing the multiple independent activities of eIF1A. The low number of affected genes was unexpected, given the strong effect seen in the polysome profiles and the metagene analysis of the Ribo-seq. This number is likely to present only those that pass the defined threshold. Clearly, many genes that were only modestly affected and were not included potentially contribute to the global effect. Analysis of the sequence context of the CDS start codons in downregulated genes indicated an enrichment of an AUG context characterized by an A at position $-3$ and C at position $+5$, a position that is not considered part of the canonical Kozak context (Fig. 4F,G). Intriguingly, in the 48S structure, this position is found very close to the NTT of eIF1A, which is the target of 1Ai-8214 and most likely indirectly modulated by 1Ai-3638 (Fig. 4H). The Ribo-seq data was also used to explore the 5'UTR features among downregulated mRNAs, which revealed a high frequency of uORFs presence in the affected gene set, suggesting a role of eIF1A in uORF regulation. This possibility was further investigated using reporter assays designed to distinguish between uORF-directed reinitiation and leaky scanning. Our findings strongly suggest that eIF1A plays an important role in the regulation of uORF-mediated re-initiation under basal conditions, especially when the re-initiation scanning distance is short. Thus, the 1Ais enabled the discovery of previously unknown mRNA features directing eIF1A dependency.

The translational downregulation of genes involved in cell cycle progression and cancer pathways, agrees with the previously reported effects of eIF1A KD (Sehrawat et al, 2019). Motivated by these observations, we studied the potential impact of the eIF1A-specific inhibitors, 1Ai-3638 and 1Ai-8214, on ovarian cancer development in xenografts. In the OVCAR8 cell-derived xenograft model, mice treated with these two 1Ais exhibited a clear reduction in tumor size and weight without apparent toxicity, suggesting that targeting eIF1A with these small molecules has the potential to decrease ovarian tumor growth. These findings are particularly relevant given the connection between eIF1A mutations and certain cancers, including ovarian carcinoma (Etemadmoghadam et al, 2017; Ewens et al, 2014; Karunamurthy et al, 2016), emphasizing the need for further exploration of their therapeutic applications.

The investigation of 1Ais as potential antiviral agents against SARS-CoV-2 revealed that several 1Ais, particularly those exclusively targeting RPS10, significantly improve the survival of infected cells, with 1Ai-4927 demonstrating greater potency. These results imply a crucial role of RPS10 interaction surface in SARS-CoV-2 successful infection. Upon exploring the underlying mechanism, we observed that 1Ais do not interfere with NSP1-mediated inhibition of translation. The RPS10-binding but not eIF1A-binding 1Ais specifically inhibit translation from a specific regulatory element in the sub-genomic 5'UTR of late genes, suggesting an eIF1A-independent function of RPS10. This element consists of a

near-cognate CUG codon bearing a specific downstream context (Havkin-Solomon et al, 2023). The inhibitory effect on CUG initiation was unexpected, given the general trend observed for 1Ais in enhancing CUG initiation (Fig. 1D; Appendix Fig. S1A), indicating that the impact of 1Ai-4927 on this specific CUG is context-dependent. Indeed, modifying this CUG resulted in the loss of sensitivity to 1Ai-4927. This is reminiscent of RPS3 mutations, which display similar differential effects on the same CUG initiations (Havkin-Solomon et al, 2023). Considering the proximity of RPS10 and RPS3 within the entry channel of the ribosome, it is likely that they cooperate in regulating this element. It is possible that the binding of 1Ai-4927 to RPS10 interferes with the mRNA binding activity of RPS10 to this CUG downstream element. These findings propose a novel route for inhibiting SARS-CoV-2 propagation by targeting translation initiation with highly specific features.

Our experiments provide compelling evidence supporting the specificity of the eIF1A-RPS10 inhibitors. First, the screening approach employs recombinant proteins, enabling the identification of compounds that directly interact with the target proteins. Indeed, most of the thoroughly investigated compounds, 1Ai-3638, 1Ai-8214, and 1Ai-4927, bind eIF1A, RPS10, or both and interfere with the association of eIF1A with the 48S ribosomal subunit in vitro and in vivo. Additionally, the rapid translational effects of these inhibitors fit the anticipated outcomes of eIF1A KD in mammalian cells. Furthermore, the translatome data reveal specific effects on a subset of mRNAs characterized by distinct translation regulatory features. This observation is substantiated by controllable reporter assays. However, it is important to note that the potential impact of these compounds on other cellular processes cannot be excluded.

In summary, our study identified the first eIF1A-RPS10 inhibitors and demonstrated their diverse effects on translation initiation and cellular processes. The use of the 1Ais not only confirmed known functions of eIF1A in mammalian cells but also uncovered novel roles for both eIF1A and RPS10. These include expansion of the AUG context, facilitating uORFs-mediated reinitiation, and modulating context-dependent upstream CUG initiation. We also demonstrated the therapeutic potential of 1Ais against cancer and viral infections. These observations underscore the importance of further investigating these small molecules and their implications for translational control and developing therapeutic applications.

## Methods

### Reagents and tools table

| Reagent/resource | Reference or source | Identifier or catalog number |
| --- | --- | --- |
| **Experimental models** | | |
| BL-21 E. coli | | |
| Dh5α E. coli | | |
| HEK293T cells | | |
| OVCAR8 | Creative Biolabs | IOC-ZP305 |
| Vero E6 | Sigma-Aldrich | 85020206 |
| Mice: C57 black | Harlan Laboratories Israel | #701 |
| Mice: NSG | Harlan Laboratories Israel | #1978 |
| **Recombinant DNA** | | |

| Reagent/resource | Reference or source | Identifier or catalog number |
| --- | --- | --- |
| PRSFduet | Merck-novagen | 71341 |
| GFP | Sehrawat et al, 2019 | |
| Firefly luciferase | | |
| Renilla luciferase | | |
| bdSumo-pet28 | Frey and Gorlich, 2014 | |
| pCRUZ-HA-NSP1 | Slobodin et al, 2022 | |
| **Antibodies** | | |
| Anti-GFP | Abcam | #ab290 |
| Anti-eIF1A | Abcam | #ab172623 |
| Anti-RPS10 | Santa Cruz | #sc-515655 |
| Anti-RPL36A | Antibody verify | #AA584660C |
| **Oligonucleotides and other sequence-based reagents** | | |
| PCR primers | This study | Table EV2 |
| qPCR primers | This study | Table EV2 |
| **Chemicals, enzymes and other reagents** | | |
| Luciferin | | |
| CTZ | | |
| Sumo protease | Frey and Gorlich, 2014 | |
| **Software** | | |
| IC50 calculator AAT Bioquest | https://www.aatbio.com/tools/ic50-calculator | |
| UTAP transcriptome analysis pipeline | Kohen et al, 2019 | |
| RNAcentral database | https://rnacentral.org/ | |
| **Other** | | |
| Monolith protein labeling kit RED-NHS 2nd generation | Nano-Temper | #MO-L011 |
| BCA protein conc. kit | Thermo Scientific | #SH255494 |
| T7 RiboMax™ | Promega | #P1280 |
| Vaccinia capping kit | New England Biolabs | #M2080S |
| Rabbit Reticulocytes lysate | Promega | #L4960 |
| PLA kit - far red | Sigma-Aldrich | #Duo92101 |
| Cell TiterGlo viability assay | Promega | #G7570 |

### Plasmids

Plasmids for the split-RL HTS were constructed on a PRSFduet backbone ((Merck-Novagen), which was inserted with EIF1A-ΔC-Renilla and RPS10-ΔN-Renilla genes from the respective mammalian expression plasmids (Haimov et al, 2017) by restriction-free (RF) cloning (Unger et al, 2010) (Primers in Appendix Table S2). For RPS10 protein expression and purification, RPS10 cDNA was PCR amplified from the above-described plasmid used for the split-RL HTS and inserted into Pet38a plasmid by RF cloning (primers in Appendix Table S2). For eIF1A protein expression and purification, eIF1A insert was PCR amplified from the above-described plasmid used for the split-RL HTS and inserted into bdSumo-Pet38a (Frey and Gorlich, 2014). The constructs of IRF7-

Kozak, IRF7/C5 and IRF7-Kozak/C5 were prepared by RF cloning based on the IRF7 firefly reporter gene. Next, a specific barcode was added to each construct after the CDS stop codon (primers in Appendix Table S2). To generate SARS-COV-2 ORF7a 5'UTR in front of the Firefly and Renilla luciferase reporters, we designed primers for RF cloning of SARS-COV-2 sub-genomic 5'UTR using primers bearing the AUG flanking nucleotide matching ORF7a in which the CUG or AUG are in frame with the reporter (primers at Appendix Table S2). Additionally, to construct the firefly luciferase reporter, in which the exclusive initiation site of the firefly luciferase is CUG, we replaced the Renilla with the firefly by RF cloning (primers at Appendix Table S2). The same firefly reporter initiating from a CUG was used as a backbone for mutations in the CUG starting codon and the AAACG regulatory element (primers in Appendix Table S2). Plasmids for testing uORFs regulation (Fig. 4L) were constructed based on the reference 5'UTR backbone (16) by RF cloning (primers at Appendix Table S2). Plasmids for testing the effect of position +5 on 1Ais inhibition of translation were constructed by mutating the IRF7 position +5, −3 and inserting a barcode after the stop codon of the firefly reporter by RF cloning (primers in Appendix Table S2).

## Cell lines maintenance and transfections

HEK293T and OVCAR8 cell lines were maintained in Dulbecco modified Eagle medium (DMEM) supplemented with 10% fetal bovine serum, 100 U/ml penicillin,and 100 g/ml streptomycin. Cells were grown at 37 °C and 5% $CO_2$. All Cells were transfected by standard $CaCl_2$ transfection in a 24-well plate with a total of 1 μg DNA per well.

## Split-Renilla assay for drug screen

eIF1A-ΔC-Renilla and RPS10-ΔN-Renilla were transformed into BL21 E. coli by heat shock transformation. A bacterial starter of 5 ml LB was incubated overnight at 37 °C and transferred into 1 L LB for incubation at 37 °C until OD ~ 0.5–0.7. Next, protein expression was induced with 0.5 mM IPTG, and bacterial cells were incubated overnight at 16 °C. Bacterial cell pellet was collected after centrifugation (3000 RPM, 15 min), resuspended in 20 ml TKE buffer (Tris 10 mM, KCl 100 mM, EDTA 0.1 mM), sonicated (bath sonicator, 24 cycles of 30 s) and centrifuged again (15000 RPM, 30 min). Supernatants were collected, analyzed for RL activity, and then used for high throughput drug screen (HTS) at the Weizmann G-INCPM HTS unit as described (Ashkenazi et al, 2017). In addition to the standard controls of this protocol, the two parts of the split RL, without eIF1A or RPS10, were tested as controls. Selected compounds were purchased from Sigma-Aldrich and Enamine. Compound powders were resuspended in DMSO for a final concentration of 10 mM and stored at −20 °C.

## Reporter assays

For the reference vs IRF7 5'UTR luciferase reporter assay, HEK239T cells in a 12-well plate were transfected with the previously described IRF7 5'UTR or reference firefly luciferase reported genes (Sehrawat et al, 2019) (100 ng/well) together with GFP (20 ng/well) that served as an internal control by CaCl2

transfection. After 6 h, half of the growth medium was replaced with a medium containing the different compounds, reaching a final concentration of 20 μM. Twenty-four hours after transfection, transfection efficiency was evaluated using GFP fluorescence intensity (Cytation 5, ex/em: 470/525). Cells were harvested in 50 μL Reported lysis buffer (RLB) and put on a shaker for 10 min. From each well 5 μL were taken in duplicates and tested by luminometer.

For the analysis of translation from very short 5'UTR and Leaky scanning, HEK293T cells were transfected with a GFP (20 ng/well) reporter gene driven by a cap-proximal and downstream AUGs (Elfakess et al, 2011) together with a luciferase reporter gene (100 ng/well) as normalizing control and treated with 20 μM of 1Ais as described above. GFP levels were determined by Western blot (WB) (GFP antibody, Abcam #ab290) and quantified by densitometry (ImageJ) and normalized to the measured luciferase activity.

For the near-cognate AUG Starting codon fidelity reporter assay, HEK293T cells were transfected with 100 ng of the previously described luciferase reporter constructs driven either by AUG, CUG, and GUG (Tang et al, 2017) together with GFP expressing plasmid (20 ng) as normalizing control. Luciferase and GFP levels were determined as described above.

For the NSP1 inhibition of translation and 1Ais, HEK293 cells were transfected with NSP1 (20 ng/well) and GFP (10 ng/well) and after 6 h 1Ais were added (20 μM). After overnight exposure, GFP was measured using cytation5. All results were normalized to cells that were not transfected with NSP1.

## Protein purification, labeling and 1Ai binding assays

His-tagged RPS10 was expressed in BL-21 E. coli strain and purified on nickel agarose beads followed by elution with 200 mM imidazole and dialysis against TKT buffer. His-sumo-tagged eIF1A was expressed in BL-21, bound to nickel agarose, and washed with TKT buffer containing 20 mM imidazole four times. eIF1A was cleaved from the His-sumo tag by incubation with the bdSumo cleavage enzyme (10 μL) on a rotator (15 RPM, 4 °C) overnight. Beads were centrifuged (1000 RPM, 1 min), and the upper liquid was collected. Protein concentration was evaluated using BCA kit (Thermo Scientific, #SH255494). Similar purifications were performed with the eIF1A NTT and CTT truncations (Appendix Table S2).

The purified RPS10 and eIF1A variants fluorescent labeling was carried outS using the Monolith protein labeling kit RED-NHS 2[nd] generation (Nano Temper, #M0-L011). Binding assays were performed in 96-well plates; 4 rows were filled with 100 μL TK buffer (for reference samples), and the other four rows were filled with 100 μL Protein (0.2 μM) in TK buffer. Four 1Ais were added to the wells of column 12 (similar 1Ais to the buffer and to the Protein) and went through serial X2 dilution (concentrations: 250 μM–0.25 μM) across the plate columns (except column 1 which was for control- no drug). After 10 min of incubation at room temperature, the plate was read in the cytation5 plate reader (ex/em - 630/690 nm). Results were calculated as a fraction of the control (no 1Ais) after subtraction of the background fluorescence in TK buffer for each drug at a specific concentration. A curve fitting was prepared and IC50 was calculated using AATBioquest IC50 calculation tool (https://www.aatbio.com/tools/ic50-calculator).

## Microscale thermophoresis

Purified eIF1A in MST buffer (300 mM KCl, 10 mM HEPES, 0.05% TWEEN20, PH 7.5) was fluorescently labeled by Monolith protein labeling kit RED-NHS 2nd generation (Nano Temper, #M0-L011). Protein was diluted to 10 ng/μL and incubated with increasing concentrations of 1Ai-3638 or 1Ai-8214 (125 μM – 2 nM). Samples were centrifuged at 15,000 RPM for 10 min at 4 °C and then loaded to MST capillaries. Initial fluorescence was measured by a Monolith 2.0 MST fluorescence reader.

## In-vitro analysis of 48S formation

mRNA was synthesized (RiboMax, #P1280, Promega) and capped (vaccinia capping kit, #M2080S, NEB) in vitro using the reference luciferase reporter DNA. Next, 50 μl rabbit reticulocyte lysate (RRL) (Promega) was incubated with 1Ais for 5 min on ice and then incubated for 5 min on ice with GMP-PNP before the addition of 2 μg mRNA and the formation of 48S ribosomal complexes. 48S complexes were formed for 30 min at 30 °C and then were subjected to sucrose gradient fractionation (8–32%). Each sucrose fraction was then precipitated by TCA and tested by WB for the different proteins (eIF1A - Abcam #ab172623, RPS10- Santa Cruz #sc-515655, RPL36A- Antibody Verify # AA584660C).

## Proximity ligation assay

Glass coverslips (13 mm, #1.5) were incubated with polylysine inside a 24-well plate for 30 min and then washed with PBS. HEK293T cells were then grown overnight on the coverslips until a high confluency (~90%) of cells was visible. Cells were treated with 1Ais for 3 h before they were washed with PBS, and 300 μl of 4% PFA was added for 20 min at room temperature. Cells were washed again (X3, PBS) and then treated with blocking and permeabilization solution (1%BSA, 5% donkey serum and 0.1% Triton in PBS) at room temperature for 1 h. Next, primary antibody treatment (eIF1A - Abcam #ab172623, RPS10- Santa Cruz #sc-515655), secondary antibody treatment, and signal amplification were performed according to Duolink PLA instructions (sigma, #Duo92101). After the final wash, each slide was incubated with DAPI (30 min, room temp), mounted on microscopy slides using Elvanol, and left overnight in the dark to dry. Fluorescence was measured by confocal microscope in three channels (Far red: ex/em: 594/624, DAPI ex/em: 354/456, and DIS contrast). For each channel, ~20 images were taken to cover all cell volumes. Images of the same channel were merged according to average light intensity and were colored (DAPI-blue, Far–red–red). Both channels of each image were then overlaid, and the brightness and contrast of all merged pictures were set to a uniform level. Nuclei and PLA dots were counted, and the ratio of dots/Nuclei was calculated for each picture. All image analysis steps were done using ImageJ.

## Ribosome footprinting and data analysis

HEK293 cells were grown until 70% confluency in a 10 cm culture dish and treated by either DMSO, 1Ai-3638 (20 μM) or 1Ai-8214 (20 μM) for 3 h. Cells were then treated with 100ug/ml CHX for 5 min and lysed in polysome buffer. After a sample was taken for the total RNA library (Mar-seq), ribosome fractions were isolated using sucrose density centrifugation and RNAseI treatment. Precipitated RNA was used for Ribo-seq library preparation as described (Ingolia et al, 2012). The final library was sequenced by Illumina Nextseq500 sequencer for SR-60.

For the data analysis, the different samples were demultiplexed according to the given index, and the adaptor sequence was removed as described (Ingolia et al, 2012). rRNA was removed by aligning sequences to rRNA database from RNAcentral (The et al, 2017) using bowtie2. Next, sequences were filtered to include only 28–32 nt long sequences using cutadapt. Sequences were aligned to human transcriptome using STAR aligner (with parameters: –outFilterMultimaoNmax 1) while transcriptome data hg38 filtered by MANEv108 (UCSC) was selected for the alignment. The number of reads per sample (CPM) was calculated for normalization. For quality control, we checked if similar samples clustered together by PCA, and samples were removed from each treatment to achieve the best clustering. Metagene plot was prepared using ribo-waltz R-package and TE for each gene was calculated by dividing each gene reads with its total RNA reads (Kohen et al, 2019). A gene was defined as downregulated if relative to DMSO $\log_2(TE) < -0.6$ and upregulated if $\log_2(TE) > 0.6$ for both samples of the same treatment. 5'UTR length and the frequency of nucleotides surrounding the main ORF AUG were determined using R Bioconductor and logos packages. The presence of uORFs in each gene's 5'UTR was determined according to previously reported data (Sehrawat et al, 2022) in which similar treatment and cellular model was used but with harringtonine treatment. This allowed the identification of initiation sites (MACS callpeak, −q 0.01) and extraction of all possible start locations for uORFs and using MACS2. We determined if the initiation site has an in-frame stop codon before the main ORF and, therefore, defined it as uORF.

For the Mar-seq data analysis, we used the UTAP transcriptome analysis pipeline (Kohen et al, 2019). The Raw reads were trimmed using cutadapt. Reads were mapped to the human genome (hg38) using STAR with the parameters –alignEndsType EndToEnd, –outFilterMismatchNoverLmax 0.05. The pipeline quantifies the 3′ of Gencode annotated genes (1000 bases upstream of the 3′ end and 100 bases downstream). UMI counting was done after marking duplicates (in-house script) using HTSeq- count in union mode. The reads with unique mapping were used for the analysis, and a minimum of five reads in at least one sample was set.

## mRNA analysis

mRNA was extracted from cells by Direct-zol RNA Mini prep kit (#R2062, ZYMO research). From each sample, 500 ng RNA was used to form cDNA (High capacity cDNA RT kit, #4368814, ThermoFisher). cDNA products were diluted 1:4 and loaded (5 μL) on a 384-wells plate in triplicates and incubated with 5 μL SYBER blue and 6 uM Primers for qPCR: Barcode 1—AGACATGAT GAACGCGCT, Barcode 2—AGACATGACAACGTGATC, Barcode 3—AGACATGATCGGATCCAG, Barcode 4—AGACATG ACCATAACGGT.

## IRF7 C + 5 mutants translation

Reporter genes were co-transfected to 10 million HEK293T cells in 10 cm plates (250 ng of each plasmid). Six hours after transfection,

cells were treated with DMSO, 1Ai-3638, or 1Ai-8214. In all, 24 h after transfection, cell lysates were prepared and ribosomal complexes were separated by sucrose gradient sedimentation. mRNA was isolated (Direct-zol RNA Mini prep, #R2062, ZYMO research) from the total fraction (before sucrose sedimentation) and the polysomes fractions. cDNA was prepared by the High-Capacity cDNA RT kit (#4368814, ThermoFisher) and then quantified by qPCR using barcoded primers. Translation efficiency values are calculated from the mRNA abundance in the polysomes fraction vs. the total in the treated groups relative to the DMSO control.

## OVCAR8 viability

In total, ~2000 OVCAR8 cells were seeded in each well of a 96-well plate (100 µL). After 6 h, 1Ai-3638 or 1Ai-8214 were added to the wells (in triplicates) in seven serial dilutions (20–0.625 µM) and 0 concentration. Every 24 h, medium with 1Ai/DMSO was refreshed, and after 72 h, cellular viability was measured using Cell TiterGlo assay (Promega #G7570).

## 1Ais toxicity in mice

In all, 21 C57/Black mice were divided into seven groups. A control group was injected with DMSO, and 6 groups were injected with either 1Ai-3638 or 1Ai-8214 at different concentrations (8/4/2 mg/kg). 1Ais was diluted to the required concentrations in PBS (According to mice weight of 20 g and injection volume 150 µL) while DMSO (1Ais solvent) concentration was 2%. All samples for injection were filtered by 0.22 µM filter. 1Ais were injected into the peritoneal cavity of the mice five times a week for 3 weeks, during which the mice's well-being was monitored, and weight was measured twice a week.

## Xenografts growth after 1Ais treatment in mice

~300 million OVCAR8 cells were grown on Petri dishes (DMEM, 10% FBS) and harvested by trypsin. Cells were then washed by PBS and diluted to 5 million cells per 150 µL in PBS. Immunocompromised (NGS) mice were treated preoperatively with one dose of buprenorphine (0.1 mg/kg) and anesthetized using 1–4% isoflurane inhalation. A small incision on the skin was made to access the ovary, and the uterine horns were traced to identify the ovary. Then, $5 \times 10^6$ OVCAR8 cells resuspended in 50 µl PBS were injected into the ovary. Mice were divided into three groups (DMSO, 1Ai-3638, and 1Ai-8214). After 3 weeks of tumor formation, mice that did not present any tumors were removed. The mice were injected with either DMSO or 8 mg/kg of 1Ais into the peritoneal cavity 5 times a week for 4 weeks before mice were sacrificed. The well-being of the animals was assessed by monitoring signs of distress, hair appearance (shiny/dirty/stiff), social behavior, signs of violence (wounds), changes in walking patterns, and stomach swelling. Immediately after mice were sacrificed, tumors were removed and weighed. Mice which did not have an ovary at all were neglected. Finally, we calculated the average tumor weight for each treatment. Tumors larger than 4 STD from the average tumor size were statistically neglected. These experiments were approved by the Institutional Animal Care and Use Committee, application number 02840322-2.

## SARS-COV2 inhibition by 1Ais in Vero E6 cells

Vero E6 cells were grown on a 96-well plate and incubated with 1Ais for 1 h before being infected with SARS-COV2 at 0.15 multiplicity of infection (MOI). 1Ais concentration was 5 µM unless we observed some toxicity to Vero E6 cells so concentration was reduced (2.5 µM – 1Ai3044 and 1Ai-5730. 1 µM – 1Ai-5175,6248 and 5662). 3 days after infection cells were tested for cytopathic effect. The final results are the percent of live cells in every treatment and control (100% is uninfected cells). Selected inhibitors were tested again at different MOI (0.1, 0.05, 0.01) and 1Ais were refreshed every 24 h ($n = 2$). The experiments were carried in Biosafety Level 2 facilities.

# Data availability

The Ribo-seq and RNA-seq datasets generated during this study have been deposited in NCBI's to the SRA database and are accessible through SRA, accession number: PRJNA1071820. Raw data of Fig. 3A,B images were uploaded to the Bio-image archive, accession number: S-BIAD1455.

The source data of this paper are collected in the following database record: biostudies:S-SCDT-10_1038-S44318-025-00449-6.

# Peer review information

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

## Acknowledgements

We would like to thank Drs. Reinat Nevo and Melanie Bokstad-Chorev for their help in the PLA imaging by the confocal microscopy; Revital Ronen from G-INCPM of the Weizmann Institute for the RNA-seq; Eliane Hadas Yardeni from the Protein Analysis Unit for her help in protein-ligand measurements. This work was supported by grants from the Minerva Foundation (#713877); Israel Science Foundation KillCorona fund (3694/20); Israel Science Foundation (#1199/22); and by Weizmann Institute internal grants from CoronaVirus Fund; the Estate of Manfred and Margaret Tannen and Joel and Mady Dukler Fund for Cancer Research. RD is the incumbent of the Ruth and Leonard Simon Chair of Cancer Research.

## Author contributions

**Daniel Hayat**: Conceptualization; Data curation; Formal analysis; Investigation; Methodology; Writing—original draft; Writing—review and editing. **Ariel Ogran**: Formal analysis; Methodology. **Shaked Ashkenazi**: Conceptualization; Data curation. **Alexander Plotnikov**: Methodology. **Roni Oren**: Methodology. **Mirie Zerbib**: Methodology. **Amir Ben-Shmuel**: Methodology. **Rivka Dikstein**: Conceptualization; Formal analysis; Supervision; Funding acquisition; Investigation; Writing—original draft; Project administration; Writing—review and editing.

Source data underlying figure panels in this paper may have individual authorship assigned. Where available, figure panel/source data authorship is listed in the following database record: biostudies:S-SCDT-10_1038-S44318-025-00449-6.

## Disclosure and competing interests statement

The authors (RD and DH) declare a patent application for the 1Ais used in this study. The remaining authors declare no competing interests.

