## [Peer Review File · The EMBO Journal]

Inhibitors of eIF1A-ribosome interaction unveil uORF-dependent regulation of translation initiation and antitumor and antiviral effects

Daniel Hayat, Ariel Ogran, Shaked Ashekenazi, Alexander Plotnikov, Roni Oren, Mirie Zerbib, Amir Ben-Shmuel, and Rivka Dikstein

Corresponding author(s): Rivka Dikstein (rivka.dikstein@weizmann.ac.il)

Review Timeline:

Submission Date:	18th Apr 24
Editorial Decision:	19th Jun 24
Appeal Received:	24th Jun 24
Editorial Decision:	6th Aug 24
Revision Received:	31st Oct 24
Editorial Decision:	13th Jan 25
Revision Received:	6th Mar 25
Accepted:	1st Apr 25

Editor: Cornelius Schneider

Transaction Report:

Dear Prof. Dikstein,

Thank you for submitting your manuscript for consideration by the EMBO Journal. It has now been seen by three referees whose comments are shown below.

As you can see from the reports that referees are in principle agree that the topic is of interest. However, all three referees also voice multiple concerns that put into question nearly all the major findings of the manuscript. Given these negative opinions and the fact that the EMBO Journal can only afford to accept papers which receive enthusiastic support from a majority of referees, I am afraid we can not offer to publish it here.

Thank you in any case for the opportunity to consider this manuscript. I am sorry we cannot be more positive on this occasion, but we hope nevertheless that you will find our referees' comments helpful.

Yours sincerely,

Cornelius Schneider

Cornelius Schneider, PhD
Editor
The EMBO Journal
c.schneider@embojournal.org

Referee #1:

In this manuscript, Hayat et al. investigated the functional consequences of the loss of eIF1A binding to the 40S ribosomal subunit by targeting the eIF1A-RPS10 interaction with help of newly identified small molecule inhibitors (1Ais). They proposed that in addition to the known roles, the eIF1A NTT-ribosome interaction also facilitates the uORF-mediated reinitiation. Moreover, they demonstrated that the eIF1A-specific 1Ais predominately affect cancer-related pathways and reduce tumor growth in xenograft models of ovarian cancer. Finally, they suggested that the RPS10-specific 1Ais may have an impact on infectivity of the SARS-Cov2 virus.

This is a clearly written, interesting story that will be valuable for the translational control field. In addition, the 1Ais identified in this work may serve as a very helpful tool to study the mechanism of proper AUG selection in higher eukaryotes. Taken together, in my opinion, this work has a merit to be published in the EMBO J., provided that the authors will resolve several important issues that I list below.

Major:

1) Changes in TE are known to affect mRNA stability. Indeed, the 1Ais affect mRNA levels in cells (Fig. 4B). Therefore, I think it is a necessity to normalize the data obtained with all reporter assays throughout the entire manuscript to mRNA levels using qPCR (at least for the 3 key 1Ais). Especially so in cases where the differences measured among various samples are relatively very small (e.g. Fig. 4F). I found the chosen normalization using the control GFP plasmid as insufficient.

In addition, for these types of reporter assays, it has almost become a standard among the fellow peers to provide a source data file (an excel file) with all raw measurements clearly organized (figure panel per sheet) and labeled. Unless I overlooked it, I would like the authors to prepare it for the next submission. Thank you.

2) Fig. 1F; have you thought of varying the Kozak sequence and the +5 base and measure their impact? I think it would be very valuable too.

3) Fig. 2C; 1Ai-4927 (binding only to Rps10) should be shown too; it could serve as a specificity control.

4) Fig. 4J - L. I think I am not 100% convinced that the authors are entitled to draw their conclusions based on measurements with only these three constructs. First of all, 16 nt is a rather short distance for the ribosome to reinitiate efficiently. Three more constructs should be added to make the conclusions convincing enough, at least for me.

A) Red and green constructs should be extended by approx. 50 nt past UAA of uORF of the red construct and measured (16+50). If the role of eIF1A in reinitiation is real, I would assume that the red50+ activity would shoot up while the green50+

would remain unchanged.

B) The green construct should be modified in such a way to make the upstream AUG codon out of frame with the downstream AUG codon, so that the upstream product overlaps with the downstream one, i.e. terminates somewhere in the downstream ORF. Comparing this construct with the existing green construct should give a definitive answer to the leaky scanning issue.

5) The RPS10 - SARS chapter is the weakest part of the manuscript. Mainly the proposed molecular mechanism of action remains obscure and I suggest either removing the reporter data from the manuscript or generating more mutant constructs to nail it down (with normalization to mRNA levels).

A) Perhaps it is my lack of knowledge but I did not really understand how CUG works here and based on what experiments the authors proposed that it works in a context-specific manner, because no context was mutated. I assume that the authors meant "AAACG". If so, it should be tested and better explained, at least for me.

B) What was also puzzling for me was the fact that 1Ai-3836 (specific for eIF1A) showed a similar effect on viability as the RPS10-specific 1Ais, yet it displayed no effect in Figure 6F.

Minor:

1) Page 3, very bottom. A lot more structural studies should be cited.

2) Fig. 1C, 1Ai-3034 increases the expression; any idea why this could be?

3) Fig. 2C. The schematic does not correspond to the text. What was first, 1Ais or mRNA?

4) Fig. 2C; It is somewhat worrisome that the RPL36A does not give any signal in the 2nd and 3rd "bottom" fractions, isn't it?

5) Page 7; this sentence breaks the flow: "Initial experiments aimed to determine eIF1A association with the 40S using cell lysates followed by sucrose gradient sedimentation or co-IP revealed that this interaction is not maintained and cannot be detected upon cell lysis."

You may think of this alternative: "Initial experiments aimed to DETECT eIF1A association with the 40S using cell lysates followed by sucrose gradient sedimentation or co-IP revealed that this interaction is not STRONG ENOUGH TO WITHSTAND THE CENTRIFUGATION FORCES."

6) The polysomes obtained with 4927 are very puzzling to me. There is barely any increase in the 80S peak compared to wt, yet the polysomes are not discernible. Any idea why this could be?

7) Fig. 4C. It is too compact, there is no way to distinguish the individual, differently color-coded peaks. For example, is there any difference on the always prominent "AUG peak" among these samples?

8) I think the relatively low number of downregulated genes (200-300) given in Fig. 4D cannot account for the relatively robust loss of polysomes seen in Fig. 4A. What are the author's thoughts about this?

9) I think Fig. 5B lacks the DMSO control.

10) Fig. 5C. The text says: "No signif. effect on the mice's well-being or any loss of weight was observed...." The loss of weight is shown but the rest is not. What else was monitored with respect to well-being? Can you be more specific?

11) Discussion, in my opinion, mostly reiterates the results without providing some insights into the molecular mechanisms underlying the effects of 1Ais. Is it all about preventing eIF1A binding to the 40S? Or can it bind with some low affinity (I assume RPS10 is not its only anchor on the ribosome) even with the drug bound and still do this while not be able to do that? And what about RPS10? What else could go wrong with this drug bound to this RPS? These kind of thoughts would be valuable to add, I think.

I thank you for the opportunity to review this article. Leos Shivaya Valasek

Referee #2:

In this manuscript, the authors investigated the function of eIF1A in translation initiation. This factor binds to the small ribosomal subunit and interacts with ribosomal protein RpS10. Using a high-throughput drug-screening approach, the authors identified specific inhibitors of eIF1A-RpS10 interaction. Three of these inhibitors were investigated more thoroughly. With these compounds in their hands, they could confirm the previously known functions of eIF1A, they also discover new roles for both RpS10 and eIF1A. More precisely, they found that the N-terminal domain (NTT) of eIF1A facilitates uORFs-mediated translation reinitiation. From a structural point of view, the structure of the 48S complex shows that the NTT is located in the vicinity of the +5 nucleotide on the mRNA (with A of the AUG being the +1). Using functional assays, they showed that a C at position +5 enhances the sensitivity of eIF1A inhibition. They used this effect to reduce cancer tumor growth in an ovarian cancer model. With dedicated reporter mRNAs, they also show that inhibition of RpS10 impacts on CUG initiation in SARS-CoV-2. Overall, this

is an interesting study that brings novel insights on the role of eIF1A although some of the brought conclusions were already known. Unfortunately, the conducted experiments are often lacking critical controls that need to be performed for conclusive interpretations.

General comments:

The order of the inhibitors is differently presented in the supplemental figures than in the main figures, this makes it difficult to compare the experiments. In addition, it would also increase the clarity of the manuscript if the 3 selected inhibitors 3638, 4927 and 8214 will be highlighted in all the figures.

Specific points:

In Figure 1C, to be conclusive on the uORF effect, these tests should also be done with a reporter containing the IRF7 5'UTR with mutated uAUGs. In Figure 1D, the authors represent the GUG/AUG initiation ratio, which might be misleading. It would be more informative to represent the values for AUG next to GUG initiation to better assess the impact of the drugs on both initiation codons. The same remarks apply to figure S1A. In addition, for both experiments, luciferase measurements should be further validated by visualization of luciferase proteins by western blots. This is critical since internal initiation might be present and might cause residual luciferase activity.

In Figure 1F, the bands corresponding to US and DS are not equally distant with the 3 inhibitors. This suggests that they are on different SDS-PAGE, it would be more informative to present the 3 inhibitors and DMSO on the same gel in order to be able to compare the intensity of each band. This would allow a direct comparison of the effect of the 3 inhibitors. One would also have an insight of the effect of the inhibitors on global translation.

In Figure 2C, the western blot analysis of 48S fractionation reveals the presence of RPL36A indicating that the fractionation between 48S and 60S is not satisfying. It is not clear if the 60S is coming for free 60S subunit or 80S. This experiment should be repeated with a more accurate fractionation and analysis of more fractions. The fractions 40S, 60S and 80S should be monitored by visualisation of 18S and 28S rRNA. In addition there is no loading control on these western blots. The presence of the introduced mRNA should also be monitored to ensure that the authors are looking at programmed-48S complex with their mRNA rather than endogenous 48S particles. In addition, this reviewer could not find the information about the amount of RNA that is used in this experiment. Indeed, the related material and method section should be more detailed.

The overlap of downregulated genes between 3638 and 8214 compounds represents 44 genes, this is a rather small proportion, 17% means that 83% do not overlap, this is puzzling with two compounds that are targeting the same molecule, namely eIF1A. How do the authors explain this result? Moreover, the ribosome profiling data should confirm the change in the uORF utilisation that was proposed in Figure 1. The authors do not comment on this at all. The impact of the inhibitors on uORF initiation should be directly visible on the ribosome profiling data. This analysis have to be included in the manuscript. Moreover, the Figure 4C which is of poor quality requires improvement to be more informative.

In Figure 4JKL, the authors use reporter Renilla proteins that contain a different N-terminal domain for -46 upstream AUG LS. It is known that the insertion of an N-terminal domain modifies the luciferase activity, therefore these constructs cannot be compared in this experiments.

In Figure 5B, the authors test the impact of the inhibitors on cell viability of OVCAR8 cells. A normal cell line should be included in this assay to ensure non-toxicity of the compounds to non-cancer cells.

In Figure 6F and H, an additional construct containing mutations of the CUG codon should be included.

In supplemental Figure 5A, the error bars are very large, a careful statistical analysis should be performed to show significant effect. According to this figure, several compounds seem to be toxic.

Minor points:

In fig1B, 29 compounds were tested in functional reporter assays, however, fig 1C and D show only 28 compounds. The text also refers to 29 compounds.

Referee #3:

In this manuscript Hayat et al describe discovery of eukaryotic translation initiation factor 1A (eIF1A) and small ribosomal protein 10 (RPS10) inhibitors. The authors fused two parts of renilla luciferase open reading frame to eIF1A and RPS10 open reading frames to generate a split renilla luciferase assay. Using this assay, authors screened ~50K compound libraries for the chemical

inhibitors of eIF1A/RPS10 interaction. These inhibitors could potentially allow for studying the role of targeted domain of a multi-functional/multidomain eIF1A in an acute setting. Authors wisely narrowed down the initial hits to 29 by screening them for inhibiting activity of intact renilla luciferase and setting 40 μ M as the minimum acceptable IC₅₀ in the split luciferase assay. Authors used these 29 compounds to determine their effect of translation of "reference" 5'UTR vs IRF7 5'UTR fused luciferase reporter or 5'UTR fused to luciferase reporters with AUG, CUG, or GUG initiation codons to dissect the effects of these compounds on translation at different context. Authors also sought to determine whether these compounds bind to a specific domain of eIF1A or interact with RPS10. Finally, authors test these compounds for inhibition of ovarian cancer cells and xenograft and reducing SARS-CoV-2 infectivity. Overall, manuscript presents a new set of compounds that have potential to be used as chemical probes to study biology of eIF1A and/or RPS10.

There are, however, several major deficiencies that prevent this reviewer from recommending the manuscript for publication in EMBO J., at least in the current form.

The most critical defect of this manuscript is the lack of experimental data to show that these set of compounds bind to eIF1A and/or RPS10, let alone data demonstrating that they are reasonably specific to eIF1A and/or RPS10. The only data presented to support binding of compounds to the eIF1A and/or RPS10 is the incubation of the compounds with fluorescent labeled eIF1A or RPS10 or their truncated forms. There is no a priori reason that compound binding should enhance or reduce the light emission from N-terminal fluorophore tagged protein. While authors incubated the compounds in the assay buffer to presumably rule out auto-fluorescence, they have not even conducted the simple experiment of incubating the compounds with the fluorophore used for labeling to rule out non-specific enhancement or quenching of the fluorophore. Some compounds appear to change fluorescent emission from every protein tested. Some compounds increase while others reduce the fluorescence light emission from the same protein in a seemingly random manner. There is no data to show whether the compounds that enhance fluorescent emission are binding or not binding the protein. Without the direct demonstration that one or more of the 29 hit compounds bind to eIF1A and/or RPS10, these compounds have limited utility; can only be used to confirm data generated by other means, i.e. replacing WT eIF1A with deletion, scanning, or point mutants. While studying biology of protein domains or subdomains by gene replacement was cumbersome before the advent of modern gene editing technology, this is no longer the case. Chemical probes whose specificity is in question are useful only as a complement to such gene replacement-based studies.

The specificity of compounds for eIF1A and/or RPS10 is not addressed et al. If we take fluorescence assay at face value, several compounds bind to both eIF1A and RPS10 and do so at lower IC₅₀ than in the luciferase assay. Given the fact that protein concentrations are much higher in fluorescence assay than in reporter assay, it is highly unlikely that the compounds' activity in the fluorescence assay is due to direct binding to the protein.

There are other major issues and some minor issues as enumerated below.

1) There is no coherence and/or correlation between the activity of hit compounds in the fluorescence assay (Figure S3B), inhibition of reporter fused to 5'UTR of IRF7 gene (1C), translation from CUG or GUG vs AUG start codons, or ratio of translation from US/DS (Figures 1D, S1A and S1B). This indicates either the assays are not robust or the observed effects are marginal. In many cases the error bars are larger than differences. Pairwise comparison by paired t-test in any form are highly inappropriate when 29 pairs are being compared. As a result, authors can detect statistical differences that has little to no biological relevance. In Figure 1F, 1Ai-3638 produces a top band distinctly different than the top band in other groups, a fact totally ignored by the authors.

2) Contrary to authors' claim, I see very little if any correlation between "binding affinity" of the hit compounds in the screening assay and fluorescent assay. Fluorescent assay IC₅₀ of every compound that allegedly bind to RPS10 is smaller than their IC₅₀ in the screening assay. In some cases, such as compound 1Ai-2383, the difference is more than 30-fold. This is not a matter of semantics but goes to the heart of problem: are these compounds specifically binding to eIF1A and/or RPS10. While it is possible that one of these compounds bind to eIF1A or RPS with some specificity, current data will not identify that compound(s). In the absence of credible affinity and specificity data it is not possible to interpret the remaining data.

3) Proximity ligation assay detects any two proteins within 40 nm. Even if the compounds were to inhibit eIF1A/RPS10 interaction, they would still be within 40 nm distance. This would be prevented only if the compounds inhibited a totally different step in the translation initiation, such as interaction of eIF4G with 40S subunit.

4) Hit compounds appear to cause accumulation of 80S mono-ribosome with limited effect on the preponderance of polyribosome at least for 1Ai-8214 and 1Ai-3638. This is reminiscent of defects in the regulation of eIF6 protein. Reduction in the ribosomal footprint in the coding region should result in increase in the ribosomal footprint in the 5'UTR. This is hardly case here in three-hour incubation period. Either these compounds do not primarily inhibit translation initiation or the authors choose an unfortunate time of incubation.

5) 1Ais appear to effect translation of a rather modest number of mRNAs even when the threshold is very modest, a 1.5-fold change. Importantly there is very little (17%) overlap in the downregulated mRNAs between 1Ai-8214 and 1Ai-3638, both of which allegedly binds to eIF1A with similar IC₅₀s and display very similar activity in the screening assay. To be very charitable to authors, their claim that 17% overlap between these two agents targeting the same protein proves their specificity is wishful thinking. Similarly, Figure 4I indicate that number of uORFs is correlated positively with probability of translational up-regulation as well as down regulation. Have the authors compared their compounds to agents that upregulate or down regulate translation of mRNAs with uORFs (i.e. BTdCPU, Chen et al, Nature Chemical Biology, 2011)?

6) Identification of C at +5 in translationally inhibited mRNAs is potentially interesting. However, as the authors must know, correlation is not causation. Affected genes also seem to have a lower frequency of G at +4. In the absence of experimental data and relatively small number of RNAs affected, it is impossible to draw any conclusion. Overall, the data in Figure 4F are difficult to interpret. What is the frequency of A vs G at -3? The sum of all nucleotides at any given position is 1. Any increase in the frequency of one nucleotide must come at the expense of one or more of the other three nucleotides at that position. In

Figure 4F, the frequency of A+G at -3 appears to be close to 0.9 in unaffected samples (by eyeballing). Yet, for affected genes, frequency of A+G at -3 appears to have doubled if not tripled. This is an artifact of presentation but in can be highly misleading. Secondary but still important points.

- 1) Authors treated Ovarian cancer xenograft bearing mice with 8 mg/kg of compounds five days a week. In my estimation 8 mg/kg will translate into ~15 uM plasma Cmax. Assuming volume of distribution equaling body weight and a well stirred single vessel model the steady state levels would be much lower unless these compound's plasma half-life exceeds 24 hours. IC50 of these compounds against the Ovcar cells used in the xenograft study is ~10 uM. The rule of thumb for anti-cancer agents is that the plasma Cmax should be 10x IC50. What is the explanation for this unexpected potency?
- 2) Authors should present pictures of all, not just one tumor. Selective data reporting is not acceptable.
- 3) Why was PLA assay not conducted in the tumor samples? Are these compounds really acting through inhibiting eIF1A/RPS10 or some other target(s).
- 4) In SARS-CoV-2 infection studies (Covid-19 is not a virus) why some of the most active compounds in other assays (1Ai-3638) have modest activity while another active compound, 1Ai-2814 is not tested?

** As a service to authors, EMBO Press provides authors with the possibility to transfer a manuscript that one journal cannot offer to publish to another EMBO publication or the open access journal Life Science Alliance launched in partnership between EMBO Press, Rockefeller University Press and Cold Spring Harbor Laboratory Press. The full manuscript and if applicable, reviewers' reports, are automatically sent to the receiving journal to allow for fast handling and a prompt decision on your manuscript. For more details of this service, and to transfer your manuscript please click on Link Not Available. **

Referee #1:

In this manuscript, Hayat et al. investigated the functional consequences of the loss of eIF1A binding to the 40S ribosomal subunit by targeting the eIF1A-RPS10 interaction with help of newly identified small molecule inhibitors (1Ais). They proposed that in addition to the known roles, the eIF1A NTT-ribosome interaction also facilitates the uORF-mediated reinitiation. Moreover, they demonstrated that the eIF1A-specific 1Ais predominately affect cancer-related pathways and reduce tumor growth in xenograft models of ovarian cancer. Finally, they suggested that the RPS10-specific 1Ais may have an impact on infectivity of the SARS-Cov2 virus.

This is a clearly written, interesting story that will be valuable for the translational control field. In addition, the 1Ais identified in this work may serve as a very helpful tool to study the mechanism of proper AUG selection in higher eukaryotes. Taken together, in my opinion, this work has a merit to be published in the EMBO J., provided that the authors will resolve several important issues that I list below.

Major:

1) Changes in TE are known to affect mRNA stability. Indeed, the 1Ais affect mRNA levels in cells (Fig. 4B). Therefore, I think it is a necessity to normalize the data obtained with all reporter assays throughout the entire manuscript to mRNA levels using qPCR (at least for the 3 key 1Ais). Especially so in cases where the differences measured among various samples are relatively very small (e.g. Fig. 4F). I found the chosen normalization using the control GFP plasmid as insufficient.

We will repeat the experiments to determine the mRNA levels by RT-qPCR as suggested.

In addition, for these types of reporter assays, it has almost become a standard among the fellow peers to provide a source data file (an excel file) with all raw measurements clearly organized (figure panel per sheet) and labeled. Unless I overlooked it, I would like the authors to prepare it for the next submission. Thank you.

The source data is ready and will be provided with the revised manuscript.

2) Fig. 1F; have you thought of varying the Kozak sequence and the +5 base and measure their impact? I think it would be very valuable too.

As suggested, we will make this variation and test it.

3) Fig. 2C; 1Ai-4927 (binding only to Rps10) should be shown too; it could serve as a specificity control.

We will carry out this experiment as suggested.

4) Fig. 4J - L. I think I am not 100% convinced that the authors are entitled to draw their conclusions based on measurements with only these three constructs. First of all, 16 nt is a rather short distance for the ribosome to reinitiate efficiently. Three more constructs should be added to make the conclusions convincing enough, at least for me.

A) Red and green constructs should be extended by approx. 50 nt past UAA of uORF of the red construct and measured (16+50). If the role of eIF1A in reinitiation is real, I

would assume that the red50+ activity would shoot up while the green50+ would remain unchanged.

We will generate the suggested additional construct.

B) The green construct should be modified in such a way to make the upstream AUG codon out of frame with the downstream AUG codon, so that the upstream product overlaps with the downstream one, i.e. terminates somewhere in the downstream ORF. Comparing this construct with the existing green construct should give a definitive answer to the leaky scanning issue.

It is important to note that there is a misunderstanding; the upstream AUG of the green construct is already out of frame with the downstream AUG.

5) The RPS10 - SARS chapter is the weakest part of the manuscript. Mainly the proposed molecular mechanism of action remains obscure and I suggest either removing the reporter data from the manuscript or generating more mutant constructs to nail it down (with normalization to mRNA levels).

The importance of the SARS-CoV-2 part lies in presenting a specific example of a translation regulatory feature regulated by RPS10, which was discovered through the use of Rps10-selective inhibitors. This will be further highlighted and clarified in the revised manuscript.

A) Perhaps it is my lack of knowledge but I did not really understand how CUG works here and based on what experiments the authors proposed that it works in a context-specific manner, because no context was mutated. I assume that the authors meant "AAACG". If so, it should be tested and better explained, at least for me.

We will elucidate further the proposed molecular mechanism as suggested.

B) What was also puzzling for me was the fact that 1Ai-3836 (specific for eIF1A) showed a similar effect on viability as the RPS10-specific 1Ais, yet it displayed no effect in Figure 6F.

I understand that this is a confusion. As can be seen in supplementary Fig.S5 (first row, middle graph), 1Ai-3638 does not improve the survival of infected cells, which is clearly distinct from the pro-survival effect seen by the RPS10-selective 1Ais. **Thus there is no discrepancy.**

Minor:

1) Page 3, very bottom. A lot more structural studies should be cited.

We will add additional citations.

2) Fig. 1C, 1Ai-3034 increases the expression; any idea why this could be?

We do not know. As this compound did not bind eIF1A or RPS10, it was not investigated further.

3) Fig. 2C. The schematic does not correspond to the text. What was first, 1Ais or mRNA?

The scheme will be corrected.

4) Fig. 2C; It is somewhat worrisome that the RPL36A does not give any signal in the 2nd and 3rd "bottom" fractions, isn't it?

This is a purely technical issue related either to the antibody quality or the strength of Rpl36 immobilization (very small protein), as Rpl36 signal is relatively very low.

5) Page 7; this sentence breaks the flow: "Initial experiments aimed to determine eIF1A

association with the 40S using cell lysates followed by sucrose gradient sedimentation or co-IP revealed that this interaction is not maintained and cannot be detected upon cell lysis."

This sentence will be revised.

You may think of this alternative: "Initial experiments aimed to DETECT eIF1A association with the 40S using cell lysates followed by sucrose gradient sedimentation or co-IP revealed that this interaction is not STRONG ENOUGH TO WITHSTAND THE CENTRIFUGATION FORCES.

The sentence will be revised as suggested.

6) The polysomes obtained with 4927 are very puzzling to me. There is barely any increase in the 80S peak compared to wt, yet the polysomes are not discernible. Any idea why this could be?

As this compound binds Rps10, we do suggest that it also may affect translation elongation. This will be further clarified in the revised MS.

7) Fig. 4C. It is too compact, there is no way to distinguish the individual, differently color-coded peaks. For example, is there any difference on the always prominent "AUG peak" among these samples?

We will revise the figure for better clarity.

8) I think the relatively low number of downregulated genes (200-300) given in Fig. 4D cannot account for the relatively robust loss of polysomes seen in Fig. 4A. What are the author's thoughts about this?

This number of genes reflects those that pass a defined threshold. Clearly, many genes that were only modestly affected and were not included potentially contribute to the global effect.

9) I think Fig. 5B lacks the DMSO control.

This is a misunderstanding, as point 0 is the DMSO control; this will be clarified in the revised MS.

10) Fig. 5C. The text says: "No signif. effect on the mice's well-being or any loss of weight was observed...." The loss of weight is shown but the rest is not. What else was monitored with respect to well-being? Can you be more specific?

We will revise this sentence for better accuracy.

11) Discussion, in my opinion, mostly reiterates the results without providing some insights into the molecular mechanisms underlying the effects of 1Ais. Is it all about preventing eIF1A binding to the 40S? Or can it bind with some low affinity (I assume RPS10 is not its only anchor on the ribosome) even with the drug bound and still do this while not be able to do that? And what about RPS10? What else could go wrong with this drug bound to this RPS? These kind of thoughts would be valuable to add, I think.

We will revise the discussion as suggested.

I thank you for the opportunity to review this article. Leos Shivaya Valasek

Referee #2:

In this manuscript, the authors investigated the function of eIF1A in translation initiation.

This factor binds to the small ribosomal subunit and interacts with ribosomal protein RpS10. Using a high-throughput drug-screening approach, the authors identified specific inhibitors of eIF1A-RpS10 interaction. Three of these inhibitors were investigated more thoroughly. With these compounds in their hands, they could confirm the previously known functions of eIF1A, they also discover new roles for both RpS10 and eIF1A. More precisely, they found that the N-terminal domain (NTT) of eIF1A facilitates uORFs-mediated translation reinitiation. From a structural point of view, the structure of the 48S complex shows that the NTT is located in the vicinity of the +5 nucleotide on the mRNA (with A of the AUG being the +1). Using functional assays, they showed that a C at position +5 enhances the sensitivity of eIF1A inhibition. They used this effect to reduce cancer tumor growth in an ovarian cancer model. With dedicated reporter mRNAs, they also show that inhibition of RpS10 impacts on CUG initiation in SARS-CoV-2. Overall, this is an interesting study that brings novel insights on the role of eIF1A although some of the brought conclusions were already known. Unfortunately, the conducted experiments are often lacking critical controls that need to be performed for conclusive interpretations.

General comments:

The order of the inhibitors is differently presented in the supplemental figures than in the main figures, this makes it difficult to compare the experiments. In addition, it would also increase the clarity of the manuscript if the 3 selected inhibitors 3638, 4927 and 8214 will be highlighted in all the figures.

We will revise the arrangements of the figures as suggested

Specific points:

In Figure 1C, to be conclusive on the uORF effect, these tests should also be done with a reporter containing the IRF7 5'UTR with mutated uAUGs.

The experiment presented in this figure is part of the screening process outlined in Figure 1B and was designed to examine which of the identified compounds affect the IRF7 5'UTR, which was previously shown to be sensitive to eIF1A knockdown. It is not known what conferred eIF1A dependency as this 5'UTR contains multiple features that include uORFs, long 5'UTR, weak AUG context as well as their combination. The involvement of eIF1A in uORF regulation was discovered in subsequent Ribo-seq studies and validated using a well-controlled reporter assay that focuses only on uORF regulation (Figure I-K). To strengthen the uORF effect, we will add an additional construct with increased distance between the uORF and the main ORF.

In Figure 1D, the authors represent the GUG/AUG initiation ratio, which might be misleading. It would be more informative to represent the values for AUG next to GUG initiation to better assess the impact of the drugs on both initiation codons. The same remarks apply to figure S1A. In addition, for both experiments, luciferase measurements should be further validated by visualization of luciferase proteins by western blots. This is critical since internal initiation might be present and might cause residual luciferase activity.

The GUG/AUG and CUG/AUG ratios are acceptable ways of presenting near-cognate

AUG activity in yeast and mammalian cells as they facilitate distinguishing between general and specific effects (see for example: <https://doi.org/10.1093/nar/gkx808>). For better clarity, in the revised MS, we will include the original luciferase values in the supplementary data.

While we understand the concern about potential internal initiation, it is important to note that we verified that any potential downstream internal initiation site can not generate luciferase activity. Below is the sequence of the firefly luciferase, including the 5'UTR, the AUG (in red), and the downstream nucleotides, which are the backbone of the NUG constructs. There is a downstream AUG (highlighted in green), but it is out of frame and is in a strong context. So if leaky scanning occurs, no luciferase protein is generated. This information will be added to the figure legend.

```
AGCTATTCCAGAAGTAGTGAGGAGGCTTTTTTGGAGGCCTAGGCTTTTGCAAAAAG  
CTTGATTCTTCTGACACAACAGTCTCGAACTTAAGCTGCAGTAGGCCACCATGGCC  
GGATCCTTCAACTTCCCTGAGCTCGAAGACGCCAAAAACATAAAGAAAGGCCCGGC  
GCCATTCTATCCTCTAGAGGATCGAACCGCTGGAGAGCAACTGCATAAG
```

Regarding the validation of luciferase measurements by western blot, this method is far less sensitive than measuring luciferase enzymatic activity. Considering the luciferase levels of the GUG and CUG are only 1-8% relative to AUG, it is not practical to use Western blotting for this purpose.

In Figure 1F, the bands corresponding to US and DS are not equally distant with the 3 inhibitors. This suggests that they are on different SDS-PAGE, it would be more informative to present the 3 inhibitors and DMSO on the same gel in order to be able to compare the intensity of each band. This would allow a direct comparison of the effect of the 3 inhibitors. One would also have an insight of the effect of the inhibitors on global translation.

Indeed, they are from different gels, each with DMSO control. In this Figure, we present only the 3 selected 1Ais with a representative DMSO, while the original gels were done with all 1Ais. For better clarity, we will include the original gels in the supplementary of the revised MS. Importantly, in this experiment, the comparison is between the upper and lower bands, which reflects the extent of leaky scanning. This ratio does not significantly change between gels.

In Figure 2C, the western blot analysis of 48S fractionation reveals the presence of RPL36A, indicating that the fractionation between 48S and 60S is not satisfying. It is not clear if the 60S is coming for free 60S subunit or 80S.

This experiment should be repeated with a more accurate fractionation and analysis of more fractions. The fractions 40S, 60S and 80S should be monitored by visualisation of 18S and 28S rRNA. In addition there is no loading control on these western blots. The presence of the introduced mRNA should also be monitored to ensure that the authors are looking at programmed-48S complex with their mRNA rather than endogenous 48S particles. In addition, this reviewer could not find the information about the amount of RNA that is used in this experiment. Indeed, the related material and method section should be more detailed.

We appreciate this comment and would like to explain. While there is some overlap

between the last fraction of the 40S and the first fraction of the 80S, this is very common (see, for example, Sinvani et al., 2015, Cell Metabolism) and depends on the fractionation method and the volume of the collected fractions. Importantly, the position of eIF1A overlaps with the 40S and is clearly distinguished from Rpl36. Furthermore, the shift of eIF1A from 40S to the top/free fractions upon drug treatment is clear. We have been using this approach for more than a decade. In the absence of mRNA, the nice 48S peak is not apparent. We will include information on the amount of the mRNA used.

The overlap of downregulated genes between 3638 and 8214 compounds represents 44 genes, this is a rather small proportion, 17% means that 83% do not overlap, this is puzzling with two compounds that are targeting the same molecule, namely eIF1A. How do the authors explain this result? Moreover, the ribosome profiling data should confirm the change in the uORF utilisation that was proposed in Figure 1. The authors do not comment on this at all. The impact of the inhibitors on uORF initiation should be directly visible on the ribosome profiling data. This analysis have to be included in the manuscript. Moreover, the Figure 4C which is of poor quality requires improvement to be more informative.

eIF1A has several functional domains, two with opposing functions (NTT and CTT). Therefore, it is expected that targeting different domains may result in somewhat different effects. Both 1Ai-8214 and 1Ai-3638 target eIF1A, but their binding sites are different. 1Ai-8214 binds NTT and 1Ai-3638 binds core RNA binding domain of eIF1A. This differential binding may have differential effect on different mRNAs. Although the number of overlapping genes is not high it is beyond the number expected by chance and is highly statistically significant (Bootstrapping, $p < 10^{-5}$). While this is explicitly explained in the discussion, we will further clarify this in the results section of the revised MS.

Regarding the impact of the inhibitors on uORF initiation, we will analyze the Ribo-seq data as suggested.

In Figure 4JKL, the authors use reporter Renilla proteins that contain a different N-terminal domain for -46 upstream AUG LS. It is known that the insertion of an N-terminal domain modifies the luciferase activity, therefore these constructs cannot be compared in this experiments.

It is important to note that there is a misunderstanding; the luciferase is not in the same frame with the upstream AUG, thus its N-terminus is not altered. This will be clarified.

In Figure 5B, the authors test the impact of the inhibitors on cell viability of OVCAR8 cells. A normal cell line should be included in this assay to ensure non-toxicity of the compounds to non-cancer cells.

These compounds affect the proliferation of cancerous (OVCAR8) as well as noncancerous cells (MEFs and HEK 293).

In Figure 6F and H, an additional construct containing mutations of the CUG codon should be included.

We will generate these constructs as suggested.

In supplemental Figure 5A, the error bars are very large, a careful statistical analysis should be performed to show significant effect. According to this figure, several compounds seem to be toxic.

The toxicity observed stems from the viral infection and is alleviated by many 1Ais. As

we have indicated in the MS, most of the compounds are indeed toxic at highest concentrations. Statistical analysis will be added.

Minor points:

In fig1B, 29 compounds were tested in functional reporter assays, however, fig 1C and D show only 28 compounds. The text also refers to 29 compounds.

This will be corrected.

Referee #3:

In this manuscript Hayat et al describe discovery of eukaryotic translation initiation factor 1A (eIF1A) and small ribosomal protein 10 (RPS10) inhibitors. The authors fused two parts of renilla luciferase open reading frame to eIF1A and RPS10 open reading frames to generate a split renilla luciferase assay. Using this assay, authors screened ~50K compound libraries for the chemical inhibitors of eIF1A/RPS10 interaction. These inhibitors could potentially allow for studying the role of targeted domain of a multi-functional/multidomain eIF1A in an acute setting. Authors wisely narrowed down the initial hits to 29 by screening them for inhibiting activity of intact renilla luciferase and setting 40 uM as the minimum acceptable IC50 in the split luciferase assay. Authors used these 29 compounds to determine their effect of translation of "reference" 5'UTR vs IRF7 5'UTR fused luciferase reporter or 5'UTR fused to luciferase reporters with AUG, CUG, or GUG initiation codons to dissect the effects of these compounds on translation at different context. Authors also sought to determine whether these compounds bind to a specific domain of eIF1A or interact with RPS10. Finally, authors test these compounds for inhibition of ovarian cancer cells and xenograft and reducing SARS-CoV-2 infectivity. Overall, manuscript presents a new set of compounds that have potential to be used as chemical probes to study biology of eIF1A and/or RPS10.

There are, however, several major deficiencies that prevent this reviewer from recommending the manuscript for publication in EMBO J., at least in the current form. The most critical defect of this manuscript is the lack of experimental data to show that these set of compounds bind to eIF1A and/or RPS10, let alone data demonstrating that they are reasonably specific to eIF1A and/or RPS10. The only data presented to support binding of compounds to the eIF1A and/or RPS10 is the incubation of the compounds with fluorescent labeled eIF1A or RPS10 or their truncated forms. There is no a priori reason that compound binding should enhance or reduce the light emission from N-terminal fluorophore tagged protein. While authors incubated the compounds in the assay buffer to presumably rule out auto-fluorescence, they have not even conducted the simple experiment of incubating the compounds with the fluorophore used for labeling to rule out non-specific enhancement or quenching of the fluorophore. Some compounds appear to change fluorescent emission from every protein tested. Some compounds increase while others reduce the fluorescence light emission from the same protein in a seemingly random manner. There is no data to show whether the compounds that enhance fluorescent emission are binding or not binding the protein. Without the direct demonstration that one or more of the 29 hit compounds bind to

eIF1A and/or RPS10, these compounds have limited utility; can only be used to confirm data generated by other means, i.e. replacing WT eIF1A with deletion, scanning, or point mutants. While studying biology of protein domains or subdomains by gene replacement was cumbersome before the advent of modern gene editing technology, this is no longer the case. Chemical probes whose specificity is in question are useful only as a complement to such gene replacement-based studies.

The specificity of compounds for eIF1A and/or RPS10 is not addressed et al. If we take fluorescence assay at face value, several compounds bind to both eIF1A and RPS10 and do so at lower IC50 than in the luciferase assay. Given the fact that protein concentrations are much higher in fluorescence assay than in reporter assay, it is highly unlikely that the compounds' activity in the fluorescence assay is due to direct binding to the protein.

There are other major issues and some minor issues as enumerated below.

We appreciate these concerns and would like to clarify the misunderstanding first: we are using a protein labeling kit that randomly attaches a fluorophore to lysine residues throughout the protein, so the protein is not just N-terminally labeled as inferred by the referee. Second, the binding assay reported here is carried out with a purified labeled protein in the presence of increasing ligand concentration. The changes in the fluorescence inform the binding state of a molecule, caused by changes in the electrostatic surrounding of the fluorescent dye when complex formation takes place. These changes in fluorescent intensity are used to calculate the dissociation constant.

To further address these concerns, we will carry out the Microscale Thermophoresis assay (MST), a well-established binding assay with the 3 major compounds studied here. This additional data will provide further assessment of the binding specificity and affinity of the compounds.

1) There is no coherence and/or correlation between the activity of hit compounds in the fluorescence assay (Figure S3B), inhibition of reporter fused to 5'UTR of IRF7 gene (1C), translation from CUG or GUG vs AUG start codons, or ratio of translation from US/DS (Figures 1D, S1A and S1B). This indicates either the assays are not robust or the observed effects are marginal. In many cases, the error bars are larger than differences. Pairwise comparison by paired t-test in any form are highly inappropriate when 29 pairs are being compared. As a result, authors can detect statistical differences that has little to no biological relevance. In Figure 1F, 1Ai-3638 produces a top band distinctly different than the top band in other groups, an fact totally ignored by the authors.

A. It is important to note that we performed a series of experimental steps outlined in the flowchart shown in Fig. 1B (also below) that aim to find compounds displaying a strong relationship between target binding and function. Specifically, 29 hits derived from the HTS were further screened for known eIF1A functions (IRF7, CUG/GUG and cap-proximal leaky scanning) and then direct binding to eIF1A and/or RPS10. This resulted in 16 compounds that bind the eIF1A and/or RPS10 and display statistically significant effect in at least one function. Of those, we then selected for further investigation the top 3 that bind the target and potentially affect 2 or 3 eIF1A-related activities (1Ai-3638, 8214, 4927). These 3 compounds have **a clear coherence between binding and function.**

Altogether, these steps strengthen relationships that may not be apparent through pairwise comparisons alone.

B. We acknowledge that in some instances, the error bars are large. However, our focus throughout the study was on changes that are reproducible and statistically significant. We will reassess our data presentation and provide further statistical tests.

C. Regarding Figure 1F, the presented data are from different gels, which may account for the slight differences in migration position. In this figure, we present only 3 selected 1Ais. The original gels, which included all 1Ais, will be provided in the supplementary materials of the revised manuscript for better clarity.

Importantly, in this experiment, the comparison refers to the ratio between the upper and lower bands, reflecting the extent of leaky scanning. In the absence of 1Ais, this ratio remains consistent across different gels, as supported by previous studies (see Elfakess et al, NAR 2011; Sinvani et al, Cell Metabolism, 2015, Haimov et al., MCB 2017 and others). The distinct top band produced by 1Ai-3638 reflects reduced cap-proximal initiation and enhanced leaky scanning.

2) Contrary to authors' claim, I see very little if any correlation between "binding affinity" of the hit compounds in the screening assay and fluorescent assay. Fluorescent assay IC50 of every compound that allegedly bind to RPS10 is smaller than their IC50 in the screening assay. In some cases, such as compound 1Ai-2383, the difference is more than 30-fold. This is not a matter of semantics but goes to the heart of problem: are these compounds specifically binding to eIF1A and/or RPS10. While it is possible that one of these compounds bind to eIF1A or RPS with some specificity, current data will not identify that compound(s). In the absence of credible affinity and specificity data it is not possible to interpret the remaining data.

We appreciate this comment and will rephrase this claim for better accuracy. It is expected that, out of the 29 compounds identified by the screen, there will be some outliers, such as 1Ai-2383, which exhibit significant discrepancies in IC50 values. Notably, this compound was not used for further studies. As indicated above, we will provide further assessment of the binding specificity and affinity of the compounds using Microscale Thermophoresis assay (MST).

3) Proximity ligation assay detects any two proteins within 40 nm. Even if the

compounds were to inhibit eIF1A/RPS10 interaction, they would still be within 40 nm distance. This would be prevented only if the compounds inhibited a totally different step in the translation initiation, such as interaction of eIF4G with 40S subunit.

Unlike eIF4G1, which is present only in the 48S pre-initiation complex, eIF1A is a component of both the 43S and 48S complexes. Our PLA experiments demonstrate that in the presence of 1Ais, the binding of eIF1A to the ribosome is not only impaired but also shifted away from RPS10 by a distance greater than 40 nm. This confirms a disruption in the eIF1A/RPS10 interaction. While it is difficult to draw conclusions from PLA regarding translation initiation stages, our findings suggest that the compounds may be affecting both the 43S and 48S stages of translation initiation. We will extend this explanation in the revised manuscript to clarify these observations.

4) Hit compounds appear to cause accumulation of 80S mono-ribosome with limited effect on the preponderance of polyribosome at least for 1Ai-8214 and 1Ai-3638. This is reminiscent of defects in the regulation of eIF6 protein. Reduction in the ribosomal footprint in the coding region should result in increase in the ribosomal footprint in the 5'UTR. This is hardly case here in three-hour incubation period. Either these compounds do not primarily inhibit translation initiation or the authors choose an unfortunate time of incubation.

This comment represents several points of misunderstanding. In polysome profiles, the translation rate is expressed as the polysome-to-monosome (P/M) ratio, which decreases with translation initiation defects and increases with defects in elongation. Our polysome profile analyses demonstrate a clear decrease in the P/M ratio in the presence of 1Ais, indicating that these compounds inhibit translation initiation (Figure 4A).

Regarding the specific compounds 1Ai-8214 and 1Ai-3638, in addition to the P/M decrease, the subsequent Ribo-seq data further supports the inhibition of translation initiation, showing a reduced ribosomal footprint in the coding region without a corresponding increase in the 5'UTR footprint, which aligns with a block at the initiation stage (Fig. 5C).

It is important to note that eIF6 is not considered a canonical translation initiation factor. While eIF6 has roles in ribosome biogenesis and regulation, the observed effects in our study are more directly related to the disruption of initiation factor interactions rather than eIF6 regulation.

In the revised manuscript, we will include clearer explanations of these observations.

5) 1Ais appear to effect translation of a rather modest number of mRNAs even when the threshold is very modest, a 1.5-fold change. Importantly there is very little (17%) overlap in the downregulated mRNAs between 1Ai-8214 and 1Ai-3638, both of which allegedly binds to eIF1A with similar IC50s and display very similar activity in the screening assay. To be very charitable to authors, their claim that 17% overlap between these two agents targeting the same protein proves their specificity is wishful thinking. Similarly, Figure 4I indicate that number of uORFs is correlated positively with probability of translational

up-regulation as well as down regulation. Have the authors compared their compounds to agents that upregulate or down regulate translation of mRNAs with uORFs (i.e. BTdCPU, Chen et al, Nature Chemical Biology, 2011)?

eIF1A has several functional domains, two with opposing functions (NTT and CTT). Therefore, it is expected that targeting different domains may result in somewhat different effects. Both 1Ai-8214 and 1Ai-3638 target eIF1A, but their binding sites are different. 1Ai-8214 binds NTT and 1Ai-3638 binds core RNA binding domain of eIF1A. This differential binding may have differential effect on different mRNAs. Although the number of overlapping genes is not high it is beyond the number expected by chance and is highly statistically significant (Bootstrapping, $p < 10^{-5}$). In addition, the overall number of genes reflects those that pass the threshold. Clearly, many genes that were only modestly affected and were not included certainly contribute to the global effect. We will expand the reference to these points in the results section of the revised MS.

We looked at the cited study (i.e. BTdCPU, Chen et al, Nature Chemical Biology, 2011). As this study does not contain Ribo-seq data, the suggested comparison can not be done. Additionally, we did not observe eIF2a-related stress pathways affected by 1Ais in our ribo-seq data, so this comparison is less relevant.

6) Identification of C at +5 in translationally inhibited mRNAs is potentially interesting. However, as the authors must know, correlation is not causation. Affected genes also seem to have a lower frequency of G at +4. In the absence of experimental data and relatively small number of RNAs affected, it is impossible to draw any conclusion. Overall, the data in Figure 4F are difficult to interpret. What is the frequency of A vs G at -3? The sum of all nucleotides at any given position is 1. Any increase in the frequency of one nucleotide must come at the expense of one or more of the other three nucleotides at that position. In Figure 4F, the frequency of A+G at -3 appears to be close to 0.9 in unaffected samples (by eyeballing). Yet, for affected genes, frequency of A+G at -3 appears to have doubled if not tripled. This is an artifact of presentation but in can be highly misleading.

We acknowledge the need for further clarification of these concerns. While the program that generates the presented LOGOs uses statistical measures and is widely used and acceptable, in the revised MS we will validate these findings by performing additional analyses and statistical measurements.

Regarding the specific values, the frequency of the A+G at the -3 position of the unaffected set is not 0.9, as pointed out by the referee, but in fact, less than 0.5 (see Figure 4F). Also, each nucleotide position is independent of the others, so an increase in a certain position is not at the expense of the other.

To avoid any potential misinterpretation and improve the understanding of the observed patterns, we will provide a more detailed examination of the nucleotide frequencies at each position and ensure that the presentation (font size) accurately reflects these values.

Secondary but still important points.

1) Authors treated Ovarian cancer xenograft bearing mice with 8 mg/kg of compounds five days a week. In my estimation 8 mg/kg will translate into ~15 uM plasma Cmax.

Assuming volume of distribution equaling body weight and a well stirred single vessel model the steady state levels would be much lower unless these compound's plasma half-life exceeds 24 hours. IC50 of these compounds against the Ovar cells used in the xenograft study is ~10 uM. The rule of thumb for anti-cancer agents is that the plasma Cmax should be 10x IC50. What is the explanation for this unexpected potency?

We appreciate this comment. The compound was administered intra-peritoneally, and not subcutaneously; thus, the local concentration near the tumor is much higher. This will be clarified in the method section.

2) Authors should present pictures of all, not just one tumor. Selective data reporting is not acceptable.

Unfortunately, we did not take pictures of all tumors, only a few representatives but all tumors were weighed and this data is presented in Fig.6D. The weight corresponded well to the size.

3) Why was PLA assay not conducted in the tumor samples? Are these compounds really acting through inhibiting eIF1A/RPS10 or some other target(s).

The animals were sacrificed 24h after the last injection, a time point by which the drug has been largely cleared from the animal. This timing precluded the possibility of conducting the PLA assay on tumor samples.

4) In SARS-CoV-2 infection studies (Covid-19 is not a virus) why some of the most active compounds in other assays (1Ai-3638) have modest activity while another active compound, 1Ai-2814 is not tested?

As indicated in the text, the most effective antiviral compounds were those that bind RPS10, not eIF1A. The active compounds in the other assays bind eIF1A, which explains the observed differences in activity. Regarding 1Ai-2814, at the time these infection experiments were done, this compound was out of stock.

Dear Prof. Dikstein,

Thank you for re-submitting your manuscript and for sharing a preliminary point-by-point response to the referee comments.

I apologize for the delay in my assessment. I reached out to several experts for an independent opinion on the concerns raised by referee #3 (and your revision plan for addressing these concerns) but unfortunately we did not receive any feedback up to now. We do not want to delay our decision any further and we also think that your preliminary point-by-point overall proposes sensible revisions. We have therefore decided to invite revisions of the manuscript based on the revisions proposed in the preliminary point-by-point response.

Concerning re-review of the revised manuscript we will share the revised manuscript with the original referees #1 and #2 but we will likely involve an arbitrating referee to assess your response to the concerns raised by the original referee #3 in order to obtain a fresh and independent opinion.

Thank you for the opportunity to consider your work for publication. I look forward to your revision.

Yours sincerely,

Cornelius Schneider

Cornelius Schneider, PhD
Editor
The EMBO Journal
c.schneider@embojournal.org

We realize that it is difficult to revise to a specific deadline. In the interest of protecting the conceptual advance provided by the work, we recommend a revision within 3 months (4th Nov 2024). Please discuss the revision progress ahead of this time with the editor if you require more time to complete the revisions. Use the link below to submit your revision:

Referee #1:

In this manuscript, Hayat et al. investigated the functional consequences of the loss of eIF1A binding to the 40S ribosomal subunit by targeting the eIF1A-RPS10 interaction with help of newly identified small molecule inhibitors (1Ais). They proposed that in addition to the known roles, the eIF1A NTT-ribosome interaction also facilitates the uORF-mediated reinitiation. Moreover, they demonstrated that the eIF1A-specific 1Ais predominately affect cancer-related pathways and reduce tumor growth in xenograft models of ovarian cancer. Finally, they suggested that the RPS10-specific 1Ais may have an impact on infectivity of the SARS-Cov2 virus.

This is a clearly written, interesting story that will be valuable for the translational control field. In addition, the 1Ais identified in this work may serve as a very helpful tool to study the mechanism of proper AUG selection in higher eukaryotes. Taken together, in my opinion, this work has a merit to be published in the EMBO J., provided that the authors will resolve several important issues that I list below.

Major:

1) Changes in TE are known to affect mRNA stability. Indeed, the 1Ais affect mRNA levels in cells (Fig. 4B). Therefore, I think it is a necessity to normalize the data obtained with all reporter assays throughout the entire manuscript to mRNA levels using qPCR (at least for the 3 key 1Ais). Especially so in cases where the differences measured among various samples are relatively very small (e.g. Fig. 4F). I found the chosen normalization using the control GFP plasmid as insufficient.

As suggested we repeated the experiments to determine the effect of the compounds on the mRNA levels by RT-qPCR (see supplementary Figure EV1C and D).

In addition, for these types of reporter assays, it has almost become a standard among the fellow peers to provide a source data file (an excel file) with all raw measurements clearly organized (figure panel per sheet) and labeled. Unless I overlooked it, I would like the authors to prepare it for the next submission. Thank you.

The source data is provided with the revised manuscript submission.

2) Fig. 1F; have you thought of varying the Kozak sequence and the +5 base and measure their impact? I think it would be very valuable too.

As suggested, we made a +5C variation in the Kozak of the short leader. As can be seen from the results below, this context improved initiation fidelity by enhancing the translation of the cap-proximal AUG (A) but caused a loss of sensitivity to the compounds (B).

We, therefore, decided to examine further the importance of the +5C position for the response to 1Ais using the IRF7 reporter gene. A C was introduced to the original weak

AUG context at the +5 position. In addition, the IRF7 was modified to a strong Kozak context without or with a +5C, as shown schematically in Figure 4I. The change in the +5C modified the N-terminal amino acid sequence and, consequently, protein stability due to the N-end rule; we, therefore, examined the effect of the compounds on these constructs by polysome profiling. Each of these reporter genes was barcoded and then co-transfected into cells, and the ratio of each mRNA in the polysome to total mRNA was determined by RT-qPCR using barcode-specific primers. The results revealed that the inhibitory effect of the compounds on IRF7 and IRF7-C5 is similar. However, both 3638 and 8214 inhibitory effects were moderately but significantly increased with the IRF7-Kozak/C5 compared to IRF7-Kozak (Figure 4I). These findings suggest that Kozak with a +5C is particularly sensitive to 1Ais, reinforcing the Ribo-seq-derived AUG context.

3) Fig. 2C; 1Ai-4927 (binding only to Rps10) should be shown too; it could serve as a specificity control.

We carried out this experiment as suggested and found that 1Ai-4927 did not change the association of eIF1A with the 48S (Figure EV4).

4) Fig. 4J - L. I think I am not 100% convinced that the authors are entitled to draw their conclusions based on measurements with only these three constructs. First of all, 16 nt is a rather short distance for the ribosome to reinitiate efficiently. Three more constructs should be added to make the conclusions convincing enough, at least for me.

A) Red and green constructs should be extended by approx. 50 nt past UAA of uORF of the red construct and measured (16+50). If the role of eIF1A in reinitiation is real, I would assume that the red50+ activity would shoot up while the green50+ would remain unchanged.

We appreciate this suggestion, and we generated the suggested additional construct by extending the distance between the uORS to the main ORF by 49 nt (Fig. 4L in the revised MS). As expected, this extension improved the efficiency of re-initiation (Figure 4M in the revised MS). Under these conditions, the inhibitory effects of 1Ai-3638 and 8214 were retained, while the effect of the less potent inhibitors 1Ai-5175 and 4700 was lost (Fig. 4N in the revised MS).

B) The green construct should be modified in such a way to make the upstream AUG codon out of frame with the downstream AUG codon, so that the upstream product overlaps with the downstream one, i.e. terminates somewhere in the downstream ORF. Comparing this construct with the existing green construct should give a definitive answer to the leaky scanning issue.

It is important to note that there is a misunderstanding; the upstream AUG of the green construct is already out of frame with the downstream AUG. This has now been better clarified in the text.

5) The RPS10 - SARS chapter is the weakest part of the manuscript. Mainly the proposed molecular mechanism of action remains obscure and I suggest either removing the reporter data from the manuscript or generating more mutant constructs to nail it down (with normalization to mRNA levels).

The importance of the SARS-CoV-2 part lies in presenting a specific example of a translation regulatory feature regulated by RPS10, which was discovered through the use of Rps10-selective inhibitors. In the revised manuscript, we added additional experiments to establish the underlying mechanism (Figure EV8).

A) Perhaps it is my lack of knowledge but I did not really understand how CUG works here and based on what experiments the authors proposed that it works in a context-specific manner, because no context was mutated. I assume that the authors meant "AAACG". If so, it should be tested and better explained, at least for me.

As suggested, we elucidated further the proposed molecular mechanism by generating three additional constructs. In the first, the CUG initiation was eliminated by modifying it to CUA. This resulted in an activity comparable to background levels (Figure EV7C). In the second, the CUG was modified to AUG, which substantially enhanced the activity as expected. In the third, the AAACG motif was mutated to TTTCG, which also diminished activity almost to the background, confirming its importance. Therefore, it was possible to examine the effect of the 1Ai on the active AUG. The results revealed that the extent of inhibition by 1Ai-4927 was significantly reduced (Fig. 6J in the revised MS). These findings confirm the importance of both CUG and AAACG sequences for the activity and the response to 1Ais.

B) What was also puzzling for me was the fact that 1Ai-3836 (specific for eIF1A) showed a similar effect on viability as the RPS10-specific 1Ais, yet it displayed no effect in Figure 6F.

I understand that this is a confusion. As can be seen in supplementary Figure EV7A (first row, middle graph), 1Ai-3638 does not improve the survival of infected cells, which is clearly distinct from the pro-survival effect seen by the RPS10-selective 1Ais. **Thus there is no discrepancy.**

Minor:

1) Page 3, very bottom. A lot more structural studies should be cited.

We added additional citations.

2) Fig. 1C, 1Ai-3034 increases the expression; any idea why this could be?

We do not know. As this compound did not bind eIF1A or RPS10, it was not investigated further.

3) Fig. 2C. The schematic does not correspond to the text. What was first, 1Ais or mRNA?

We thank the referee for this comment. The scheme was corrected.

4) Fig. 2C; It is somewhat worrisome that the RPL36A does not give any signal in the 2nd and 3rd "bottom" fractions, isn't it?

This is a purely technical issue related either to the antibody quality or the strength of Rpl36 immobilization (very small protein), as Rpl36 signal is relatively very low.

5) Page 7; this sentence breaks the flow: "Initial experiments aimed to determine eIF1A association with the 40S using cell lysates followed by sucrose gradient sedimentation or co-IP revealed that this interaction is not maintained and cannot be detected upon cell lysis."

You may think of this alternative: "Initial experiments aimed to DETECT eIF1A association with the 40S using cell lysates followed by sucrose gradient sedimentation or co-IP revealed that this interaction is not STRONG ENOUGH TO WITHSTAND THE

CENTRIFUGATION FORCES.

We have revised this sentence as suggested to improve clarity.

6) The polysomes obtained with 4927 are very puzzling to me. There is barely any increase in the 80S peak compared to wt, yet the polysomes are not discernible. Any idea why this could be?

As this compound binds Rps10, we think it may also affect translation elongation. This suggestion is now included.

7) Fig. 4C. It is too compact, there is no way to distinguish the individual, differently color-coded peaks. For example, is there any difference on the always prominent "AUG peak" among these samples?

We have revised the figure for better clarity.

8) I think the relatively low number of downregulated genes (200-300) given in Fig. 4D cannot account for the relatively robust loss of polysomes seen in Fig. 4A. What are the author's thoughts about this?

This number of genes reflects those that pass a defined threshold. Clearly, many genes that were only modestly affected and were not included potentially contribute to the global effect. This point was added to the discussion (second paragraph).

9) I think Fig. 5B lacks the DMSO control.

This is a misunderstanding, as point 0 is the DMSO control; this has been clarified in the revised MS.

10) Fig. 5C. The text says: "No signif. effect on the mice's well-being or any loss of weight was observed...." The loss of weight is shown but the rest is not. What else was monitored with respect to well-being? Can you be more specific?

As suggested, we added a description of the well-being assessment of the animals to the methods, which includes signs of distress, hair appearance (shiny/dirty/stiff), social behavior, signs of violence (wounds), changes in walking patterns, and stomach swelling.

11) Discussion, in my opinion, mostly reiterates the results without providing some insights into the molecular mechanisms underlying the effects of 1Ais. Is it all about preventing eIF1A binding to the 40S? Or can it bind with some low affinity (I assume RPS10 is not its only anchor on the ribosome) even with the drug bound and still do this while not be able to do that? And what about RPS10? What else could go wrong with this drug bound to this RPS? These kind of thoughts would be valuable to add, I think.

We have addressed the issue of 1Ais off-targets in the paragraph before last in the discussion as suggested.

I thank you for the opportunity to review this article. Leos Shivaya Valasek

Referee #2:

In this manuscript, the authors investigated the function of eIF1A in translation initiation. This factor binds to the small ribosomal subunit and interacts with ribosomal protein

RpS10. Using a high-throughput drug-screening approach, the authors identified specific inhibitors of eIF1A-RpS10 interaction. Three of these inhibitors were investigated more thoroughly. With these compounds in their hands, they could confirm the previously known functions of eIF1A, they also discover new roles for both RpS10 and eIF1A. More precisely, they found that the N-terminal domain (NTT) of eIF1A facilitates uORFs-mediated translation reinitiation. From a structural point of view, the structure of the 48S complex shows that the NTT is located in the vicinity of the +5 nucleotide on the mRNA (with A of the AUG being the +1). Using functional assays, they showed that a C at position +5 enhances the sensitivity of eIF1A inhibition. They used this effect to reduce cancer tumor growth in an ovarian cancer model. With dedicated reporter mRNAs, they also show that inhibition of RpS10 impacts on CUG initiation in SARS-CoV-2. Overall, this is an interesting study that brings novel insights on the role of eIF1A although some of the brought conclusions were already known. Unfortunately, the conducted experiments are often lacking critical controls that need to be performed for conclusive interpretations.

General comments:

The order of the inhibitors is differently presented in the supplemental figures than in the main figures, this makes it difficult to compare the experiments. In addition, it would also increase the clarity of the manuscript if the 3 selected inhibitors 3638, 4927 and 8214 will be highlighted in all the figures.

We have revised the arrangements of the figures as suggested.

Specific points:

In Figure 1C, to be conclusive on the uORF effect, these tests should also be done with a reporter containing the IRF7 5'UTR with mutated uAUGs.

The experiment presented in this figure is part of the screening process outlined in Figure 1B and was designed to examine which of the identified compounds affect the IRF7 5'UTR, which was previously shown to be sensitive to eIF1A knockdown. It is not known what conferred eIF1A dependency as this 5'UTR contains multiple features that include uORFs, long 5'UTR, weak AUG context as well as their combination. The involvement of eIF1A in uORF regulation was discovered in subsequent Ribo-seq studies and validated using a well-controlled reporter assay that focuses only on uORF regulation (Figure 4L-M).

To strengthen the uORF effect, we have added an additional construct with extended distance between the uORF and the main ORF by 49 nt (Fig. 4L in the revised MS). As expected, this extension improved the efficiency of re-initiation (Figure 4M in the revised MS). Under these conditions, the inhibitory effects of 1Ai-3638 and 8214 were retained, while the effect of the less potent inhibitors 1Ai-5175 and 4700 was lost (Fig. 4N in the revised MS).

In Figure 1D, the authors represent the GUG/AUG initiation ratio, which might be misleading. It would be more informative to represent the values for AUG next to GUG initiation to better assess the impact of the drugs on both initiation codons. The same

remarks apply to figure S1A. In addition, for both experiments, luciferase measurements should be further validated by visualization of luciferase proteins by western blots. This is critical since internal initiation might be present and might cause residual luciferase activity.

The GUG/AUG and CUG/AUG ratios are acceptable ways of presenting near-cognate AUG activity in yeast and mammalian cells as they facilitate distinguishing between general and specific effects (see for example: <https://doi.org/10.1093/nar/gkx808>). For better clarity, in the revised MS, we included the original luciferase values in the source data file.

While we understand the concern about potential internal initiation, it is important to note that we verified that any potential downstream internal initiation site can not generate luciferase activity. Below is the sequence of the firefly luciferase, including the 5'UTR, the AUG (in red), and the downstream nucleotides, which are the backbone of the NUG constructs. There is a downstream AUG (highlighted in green), but it is out of frame and is in a strong context. So if leaky scanning occurs, no luciferase protein is generated.

```
AGCTATTCCAGAAGTAGTGAGGAGGCTTTTTTGGAGGCCTAGGCTTTTGCAAAAAG  
CTTGATTCTTCTGACACAACAGTCTCGAACTTAAGCTGCAGTAGGCCACCATGGCC  
GGATCCTTCAACTTCCCTGAGCTCGAAGACGCCAAAAACATAAAGAAAGGCCCGGC  
GCCATTCTATCCTCTAGAGGATCGAACCGCTGGAGAGCAACTGCATAAG
```

This information has been added to the figure legend.

Regarding validating luciferase measurements by western blot, this method is far less sensitive than measuring luciferase enzymatic activity. Considering the luciferase levels of the GUG and CUG are only 1-8% relative to AUG, it is not practical to use Western blotting for this purpose.

In Figure 1F, the bands corresponding to US and DS are not equally distant with the 3 inhibitors. This suggests that they are on different SDS-PAGE, it would be more informative to present the 3 inhibitors and DMSO on the same gel in order to be able to compare the intensity of each band. This would allow a direct comparison of the effect of the 3 inhibitors. One would also have an insight of the effect of the inhibitors on global translation.

Indeed, they are from different gels, each with DMSO control. In this Figure, we present only the 3 selected 1Ais with a representative DMSO, while the original gels were done with all 1Ais. For better clarity, we included the original gels in the source data file of the revised MS. Importantly, in this experiment, the comparison is between the upper and lower bands, which reflects the extent of leaky scanning. This ratio does not significantly change between gels.

In Figure 2C, the western blot analysis of 48S fractionation reveals the presence of RPL36A, indicating that the fractionation between 48S and 60S is not satisfying. It is not clear if the 60S is coming for free 60S subunit or 80S.

This experiment should be repeated with a more accurate fractionation and analysis of more fractions. The fractions 40S, 60S and 80S should be monitored by visualisation of

18S and 28S rRNA. In addition there is no loading control on these western blots. The presence of the introduced mRNA should also be monitored to ensure that the authors are looking at programmed-48S complex with their mRNA rather than endogenous 48S particles. In addition, this reviewer could not find the information about the amount of RNA that is used in this experiment. Indeed, the related material and method section should be more detailed.

We appreciate this comment and would like to explain. While there is some overlap between the last fraction of the 40S and the first fraction of the 80S, this is very common (see, for example, Sinvani et al., 2015, Cell Metabolism) and depends on the fractionation method and the volume of the collected fractions. Importantly, the position of eIF1A overlaps with the 40S and is clearly distinguished from Rpl36. Furthermore, the shift of eIF1A from 40S to the top/free fractions upon drug treatment is clear. We have been using this approach for more than a decade. In the absence of mRNA, the nice 48S peak is not apparent. We added include information on the amount of the mRNA used to the methods section.

The overlap of downregulated genes between 3638 and 8214 compounds represents 44 genes, this is a rather small proportion, 17% means that 83% do not overlap, this is puzzling with two compounds that are targeting the same molecule, namely eIF1A. How do the authors explain this result? Moreover, the ribosome profiling data should confirm the change in the uORF utilisation that was proposed in Figure 1. The authors do not comment on this at all. The impact of the inhibitors on uORF initiation should be directly visible on the ribosome profiling data. This analysis have to be included in the manuscript. Moreover, the Figure 4C which is of poor quality requires improvement to be more informative.

eIF1A has several functional domains, two with opposing functions (NTT and CTT). Therefore, it is expected that targeting different domains may result in somewhat different effects. Both 1Ai-8214 and 1Ai-3638 target eIF1A, but their binding sites are different. 1Ai-8214 binds NTT and 1Ai-3638 binds core RNA binding domain of eIF1A. This differential binding may have differential effect on different mRNAs. Although the number of overlapping genes is not high it is beyond the number expected by chance and is highly statistically significant (Bootstrapping, $p < 10^{-5}$). We have now discussed this issue in the discussion section of the revised MS (third paragraph).

We thank the referee for the suggestion to analyze the impact of the inhibitors on uORF initiation. Upon analysis of the data, we noticed that the main difference is the first nucleotide of the initiation site. In the unaffected mRNAs, it is NUG, while in the downregulated mRNAs, it is enriched with A/CUG. This data is shown in Figure EV5D.

We have improved the quality of Figure 4C, as suggested.

In Figure 4JKL, the authors use reporter Renilla proteins that contain a different N-terminal domain for -46 upstream AUG LS. It is known that the insertion of an N-terminal domain modifies the luciferase activity, therefore these constructs cannot be compared in this experiments.

It is important to note that there is a misunderstanding; the luciferase is not in the same frame with the upstream AUG, thus its N-terminus is not altered. This has now been

further clarified.

In Figure 5B, the authors test the impact of the inhibitors on cell viability of OVCAR8 cells. A normal cell line should be included in this assay to ensure non-toxicity of the compounds to non-cancer cells.

These compounds affect the proliferation of cancerous (OVCAR8) as well as noncancerous cells (MEFs and HEK 293).

In Figure 6F and H, an additional construct containing mutations of the CUG codon should be included.

As suggested, we generated this mutation as well as two additional mutations. In the first, the CUG initiation was eliminated by modifying it to CUA. This resulted in an activity comparable to background levels (Figure EV7C). In the second, the CUG was modified to AUG, which substantially enhanced the activity as expected. In the third, the AAACG motif was mutated to TTTCG, which also diminished activity almost to the background, confirming its importance. Therefore, it was possible to examine the effect of the 1Ai on the active AUG. The results revealed that the extent of inhibition by 1Ai-4927 was significantly reduced (Fig. 6J in the revised MS). These findings confirm the importance of both CUG and AAACG sequences for the activity and the response to 1Ais.

In supplemental Figure 5A, the error bars are very large, a careful statistical analysis should be performed to show significant effect. According to this figure, several compounds seem to be toxic.

The toxicity observed stems from the viral infection and is alleviated by many 1Ais. As we have indicated in the MS, most of the compounds are indeed toxic at the highest concentrations. As suggested, we added statistical significance to each data point.

Minor points:

In fig1B, 29 compounds were tested in functional reporter assays, however, fig 1C and D show only 28 compounds. The text also refers to 29 compounds.

This has been corrected.

Referee #3:

In this manuscript Hayat et al describe discovery of eukaryotic translation initiation factor 1A (eIF1A) and small ribosomal protein 10 (RPS10) inhibitors. The authors fused two parts of renilla luciferase open reading frame to eIF1A and RPS10 open reading frames to generate a split renilla luciferase assay. Using this assay, authors screened ~50K compound libraries for the chemical inhibitors of eIF1A/RPS10 interaction. These inhibitors could potentially allow for studying the role of targeted domain of a multi-functional/multidomain eIF1A in an acute setting. Authors wisely narrowed down the initial hits to 29 by screening them for inhibiting activity of intact renilla luciferase and setting 40 uM as the minimum acceptable IC50 in the split luciferase assay. Authors used these 29 compounds to determine their effect of translation of "reference" 5'UTR

vs IRF7 5'UTR fused luciferase reporter or 5'UTR fused to luciferase reporters with AUG, CUG, or GUG initiation codons to dissect the effects of these compounds on translation at different context. Authors also sought to determine whether these compounds bind to a specific domain of eIF1A or interact with RPS10. Finally, authors test these compounds for inhibition of ovarian cancer cells and xenograft and reducing SARS-CoV-2 infectivity. Overall, manuscript presents a new set of compounds that have potential to be used as chemical probes to study biology of eIF1A and/or RPS10.

There are, however, several major deficiencies that prevent this reviewer from recommending the manuscript for publication in EMBO J., at least in the current form. The most critical defect of this manuscript is the lack of experimental data to show that these set of compounds bind to eIF1A and/or RPS10, let alone data demonstrating that they are reasonably specific to eIF1A and/or RPS10. The only data presented to support binding of compounds to the eIF1A and/or RPS10 is the incubation of the compounds with fluorescent labeled eIF1A or RPS10 or their truncated forms. There is no a priori reason that compound binding should enhance or reduce the light emission from N-terminal fluorophore tagged protein. While authors incubated the compounds in the assay buffer to presumably rule out auto-fluorescence, they have not even conducted the simple experiment of incubating the compounds with the fluorophore used for labeling to rule out non-specific enhancement or quenching of the fluorophore. Some compounds appear to change fluorescent emission from every protein tested. Some compounds increase while others reduce the fluorescence light emission from the same protein in a seemingly random manner. There is no data to show whether the compounds that enhance fluorescent emission are binding or not binding the protein. Without the direct demonstration that one or more of the 29 hit compounds bind to eIF1A and/or RPS10, these compounds have limited utility; can only be used to confirm data generated by other means, i.e. replacing WT eIF1A with deletion, scanning, or point mutants. While studying biology of protein domains or subdomains by gene replacement was cumbersome before the advent of modern gene editing technology, this is no longer the case. Chemical probes whose specificity is in question are useful only as a complement to such gene replacement-based studies.

The specificity of compounds for eIF1A and/or RPS10 is not addressed et al. If we take fluorescence assay at face value, several compounds bind to both eIF1A and RPS10 and do so at lower IC50 than in the luciferase assay. Given the fact that protein concentrations are much higher in fluorescence assay than in reporter assay, it is highly unlikely that the compounds' activity in the fluorescence assay is due to direct binding to the protein.

There are other major issues and some minor issues as enumerated below.

We appreciate these concerns and would like to clarify the misunderstanding first: we are using a protein labeling kit that randomly attaches a fluorophore to lysine residues throughout the protein, so the protein is not just N-terminally labeled as inferred by the referee. Second, the binding assay reported here is carried out with a purified labeled protein in the presence of increasing ligand concentration. The changes in the fluorescence inform the binding state of a molecule, caused by changes in the electrostatic surrounding of the fluorescent dye when complex formation takes place. These changes in fluorescent intensity are used to calculate the dissociation constant.

To further address these concerns, we carried out additional binding experiments. We repeated the purification and labeling of eIF1A and used Microscale Thermophoresis assay (MST), a well-established ligand-protein binding assay, with 1Ai-8214 and 3638. As a control for specificity, the same binding assay was done with a denatured protein (4M urea). The binding curves and IC-50 were highly similar to the original binding data, and no binding was detected when eIF1A was denatured (Figure EV2D). As Rps10 was insoluble in the MST buffer, we repeated the same fluorescence binding with 1Ai-4927 using native and denatured Rps10. Here, too, binding was detected with the native but not denatured protein (Figure EV3B). This additional data provides further validation of the binding specificity and affinity of the compounds.

1) There is no coherence and/or correlation between the activity of hit compounds in the fluorescence assay (Figure S3B), inhibition of reporter fused to 5'UTR of IRF7 gene (1C), translation from CUG or GUG vs AUG start codons, or ratio of translation from US/DS (Figures 1D , S1A and S1B). This indicates either the assays are not robust or the observed effects are marginal. In many cases, the error bars are larger than differences. Pairwise comparison by paired t-test in any form are highly inappropriate when 29 pairs are being compared. As a result, authors can detect statistical differences that has little to no biological relevance. In Figure 1F, 1Ai-3638 produces a top band distinctly different than the top band in other groups, an fact totally ignored by the authors.

A. It is important to note that we performed a series of experimental steps outlined in the flowchart shown in Fig. 1B (also below) that aim to find compounds displaying a strong relationship between target binding and function. Specifically, 29 hits derived from the HTS were further screened for known eIF1A functions (IRF7, CUG/GUG and cap-proximal leaky scanning) and then direct binding to eIF1A and/or RPS10. This resulted in 16 compounds that bind the eIF1A and/or RPS10 and display statistically significant effect in at least one function. Of those, we then selected for further investigation the top 3 that bind the target and potentially affect 2 or 3 eIF1A-related activities (1Ai-3638, 8214, 4927). These 3 compounds have **a clear coherence between binding and function (see Figures 4 and 6)**. Altogether, these steps strengthen relationships that may not be apparent through pairwise comparisons alone.

B. We acknowledge that, in some instances, the error bars are large. However, our focus throughout the study was on changes that are reproducible and statistically significant. We reassessed our data presentation and provided further statistical tests (see Figure EV7A). Importantly, all of our conclusions are from data that passed statistical analysis.

C. Regarding Figure 1F, the presented data are from different gels, which may account for the slight differences in migration position. In this figure, we present only 3 selected 1Ais. The original gels, which include all 1Ais, are provided in the source data file of the revised manuscript.

Importantly, in this experiment, the comparison refers to the ratio between the upper and lower bands, reflecting the extent of leaky scanning. In the absence of 1Ais, this ratio remains consistent across different gels, as supported by previous studies (see Elfakess et al, NAR 2011; Sinvani et al, Cell Metabolism, 2015, Haimov et al., MCB 2017 and others). The distinct top band produced by 1Ai-3638 reflects reduced cap-proximal initiation and enhanced leaky scanning.

2) Contrary to authors' claim, I see very little if any correlation between "binding affinity" of the hit compounds in the screening assay and fluorescent assay. Fluorescent assay IC₅₀ of every compound that allegedly bind to RPS10 is smaller than their IC₅₀ in the screening assay. In some cases, such as compound 1Ai-2383, the difference is more than 30-fold. This is not a matter of semantics but goes to the heart of problem: are these compounds specifically binding to eIF1A and/or RPS10. While it is possible that one of these compounds bind to eIF1A or RPS with some specificity, current data will not identify that compound(s). In the absence of credible affinity and specificity data it is not possible to interpret the remaining data.

We appreciate this comment and have rephrased this claim for better accuracy. It is expected that, out of the 29 compounds identified by the screen, there will be some outliers, such as 1Ai-2383, 5576 and 7688, which exhibit significant discrepancies in IC₅₀ values. Notably, these compounds were not used for further studies. As indicated above, we now provided a further assessment of the binding specificity and affinity of the compounds.

3) Proximity ligation assay detects any two proteins within 40 nm. Even if the compounds were to inhibit eIF1A/RPS10 interaction, they would still be within 40 nm distance. This would be prevented only if the compounds inhibited a totally different step in the translation initiation, such as interaction of eIF4G with 40S subunit.

Unlike eIF4G1, which is present only in the 48S pre-initiation complex, eIF1A is a component of both the 43S and 48S complexes. Our PLA experiments demonstrate that in the presence of 1Ais, the binding of eIF1A to the ribosome is not only impaired but also shifted away from RPS10 by a distance greater than 40 nm. This confirms a disruption in the eIF1A/RPS10 interaction. However, it is difficult to draw conclusions

from PLA regarding the 43S and 48S stages of translation initiation. We now extended this explanation in the discussion of the revised manuscript.

4) Hit compounds appear to cause accumulation of 80S mono-ribosome with limited effect on the preponderance of polyribosome at least for 1Ai-8214 and 1Ai-3638. This is reminiscent of defects in the regulation of eIF6 protein. Reduction in the ribosomal footprint in the coding region should result in increase in the ribosomal footprint in the 5'UTR. This is hardly the case here in three-hour incubation period. Either these compounds do not primarily inhibit translation initiation or the authors choose an unfortunate time of incubation.

This comment represents several points of misunderstanding. In polysome profiles, the translation rate is expressed as the polysome-to-monosome (P/M) ratio, which decreases with translation initiation defects and increases with defects in elongation. Our polysome profile analyses demonstrate a clear decrease in the P/M ratio in the presence of the 3 1Ais, indicating that these compounds inhibit translation initiation (Figure 4A).

Regarding the specific compounds 1Ai-8214 and 1Ai-3638, in addition to the P/M decrease, the subsequent Ribo-seq data further supports the inhibition of translation initiation, showing a reduced ribosomal footprint in the coding region without a corresponding increase in the 5'UTR footprint, which aligns with a block at the initiation stage (Figure 4C).

It is important to note that eIF6 is not considered a canonical translation initiation factor. While eIF6 has roles in ribosome biogenesis and regulation, the observed effects in our study are more directly related to the disruption of initiation factor interactions rather than eIF6 regulation.

5) 1Ais appear to effect translation of a rather modest number of mRNAs even when the threshold is very modest, a 1.5-fold change. Importantly there is very little (17%) overlap in the downregulated mRNAs between 1Ai-8214 and 1Ai-3638, both of which allegedly binds to eIF1A with similar IC50s and display very similar activity in the screening assay. To be very charitable to authors, their claim that 17% overlap between these two agents targeting the same protein proves their specificity is wishful thinking. Similarly, Figure 4I indicate that number of uORFs is correlated positively with probability of translational up-regulation as well as down regulation. Have the authors compared their compounds to agents that upregulate or down regulate translation of mRNAs with uORFs (i.e. BTdCPU, Chen et al, Nature Chemical Biology, 2011)?

eIF1A has several functional domains, two with opposing functions (NTT and CTT). Therefore, it is expected that targeting different domains may result in somewhat different effects. Both 1Ai-8214 and 1Ai-3638 target eIF1A, but their binding sites are different. 1Ai-8214 binds NTT and 1Ai-3638 binds core RNA binding domain of eIF1A. This differential binding may have differential effect on different mRNAs. Although the number of overlapping genes is not high, it is beyond the number expected by chance and is highly statistically significant (Bootstrapping, $p < 10^{-5}$). In addition, the overall number of genes reflects those that pass the threshold. Clearly, many genes that were

only modestly affected and were not included certainly contribute to the global effect. We refer to these points in the discussion section of the revised MS (highlighted text).

We looked at the cited study (i.e. BTdCPU, Chen et al, Nature Chemical Biology, 2011). As this study does not contain Ribo-seq data, the suggested comparison can not be done. Additionally, we did not observe eIF2a-related stress pathways affected by 1Ais in our ribo-seq data, so this comparison is less relevant.

6) Identification of C at +5 in translationally inhibited mRNAs is potentially interesting. However, as the authors must know, correlation is not causation. Affected genes also seem to have a lower frequency of G at +4. In the absence of experimental data and relatively small number of RNAs affected, it is impossible to draw any conclusion. Overall, the data in Figure 4F are difficult to interpret. What is the frequency of A vs G at -3? The sum of all nucleotides at any given position is 1. Any increase in the frequency of one nucleotide must come at the expense of one or more of the other three nucleotides at that position. In Figure 4F, the frequency of A+G at -3 appears to be close to 0.9 in unaffected samples (by eyeballing). Yet, for affected genes, frequency of A+G at -3 appears to have doubled if not tripled. This is an artifact of presentation but in can be highly misleading.

We acknowledge the need for further clarification of these concerns. Regarding the frequency, each nucleotide position is independent of the others, so an increase in a certain position is not at the expense of the other.

While the program that generates the presented LOGOs uses statistical measures and is widely used and acceptable, in the revised MS we validated these findings by performing additional analyses and statistical measurements (Figure 4G). In this analysis, the Y axis is represent the frequency.

Secondary but still important points.

1) Authors treated Ovarian cancer xenograft bearing mice with 8 mg/kg of compounds five days a week. In my estimation 8 mg/kg will translate into ~15 uM plasma Cmax. Assuming volume of distribution equaling body weight and a well stirred single vessel model the steady state levels would be much lower unless these compound's plasma half-life exceeds 24 hours. IC50 of these compounds against the Ovar cells used in the xenograft study is ~10 uM. The rule of thumb for anti-cancer agents is that the plasma Cmax should be 10x IC50. What is the explanation for this unexpected potency?

We appreciate this comment. The compounds were administrated intra-peritoneally, and not subcutaneously; thus, the local concentration near the tumor is much higher. This has now been clarified in the method section.

2) Authors should present pictures of all, not just one tumor. Selective data reporting is not acceptable.

Unfortunately, we did not take pictures of all tumors, only a few representatives but all tumors were weighed and this data is presented in Figure 6D. The weight corresponded well to the size. To avoid misunderstanding, these pictures were removed.

3) Why was PLA assay not conducted in the tumor samples? Are these compounds really acting through inhibiting eIF1A/RPS10 or some other target(s).

This is a very good point, but in the case of the xenograft study, the animals were sacrificed 24h after the last injection, a time point by which the drug has been largely cleared from the animal. This timing precluded the possibility of conducting the PLA assay on tumor samples.

4) In SARS-CoV-2 infection studies (Covid-19 is not a virus) why some of the most active compounds in other assays (1Ai-3638) have modest activity while another active compound, 1Ai-2814 is not tested?

As indicated in the text, the most effective antiviral compounds were those that bind RPS10, not eIF1A which include 1Ai-3628 and 8214. The active compounds in the other assays bind eIF1A, which explains the observed differences in activity. Regarding 1Ai-2814, at the time these infection experiments were done, this compound was out of stock.

Dear Prof. Dikstein,

Thank you for submitting a revised version of your manuscript. Your study has now been seen by all original referees, who find that many of their previous concerns have been addressed. However referees #1 and #2 still voice several concerns. We think that both the textual additions asked by referee #1 and #2 are fair and productive and we also think that the additional control experiment asked by referee #2 for figure 2c is reasonable. We would therefore ask you incorporate these minor revisions and submit a revised manuscript.

In addition we have compiled a number of mainly editorial points that have to be addressed before acceptance of the manuscript:

- During our routine pre-acceptance checks, our data editors have raised the following queries regarding figures, data, and legends:

Figure Legends (main + EV):

1. Please note that the figure titles for figures EV 1; EV 3; EV 7.
2. Please note that the figure 2d is not provided in the manuscript, however the legend for the same is provided.
3. Please define the annotated p values **/* as well as provide the exact p-values for the same in the legend of figure 5d; as appropriate.
4. Please note that the exact p values are not provided in the legends of figures 1c-d, f; 3c; 4b, g, i, m-n; 6a, f; EV 1a-b; EV 7a; EV 8b-c.
5. Please indicate the statistical test used for data analysis in the legends of figures 5d; EV 7a.
6. Please note that for the figure 6b, p-values are indicated in the legend. However, comparison for the same, ""**/*"" has not been represented in the figure. Also, provide exact p-value and statistical test for the same.
8. Please note that the scale bar is missing for figures 3a-b.
9. Please note that axis gaps are not labeled appropriately in figure EV 8b.
10. There are only n=2 replicates in figures 4b; 6b and d. Please either add additional replicates or remove the statistical analysis from the figure, legends and text and display the individual datapoints in the figure.
11. Although 'n' is provided, please describe the nature of entity for 'n' in the legends of figures 1c-d, f; 3c; 4b, i, m; 5b; 6a-b, f, h; EV 1a-d; EV 7a-b; EV 8b-c.
Authors' response: "As for your point #4, please note that for each of these sections, the nature of the entity is already described in the legends."
Editor: Please specify if the indicated n corresponds to independent biological replicates.
12. Please note that the measure of center for the error bars needs to be defined in the legends of figures 4b; 6a-b.

Additional editorial notes:

- On the abstract page of the manuscript, please include 4-5 general keyword terms to enhance searchability.
- Please rename the Conflict of Interest section into "Disclosure and Competing Interests Statement", in accordance with our updated Guide to Authors (<https://www.embopress.org/competing-interests>)
- As we are switching from a free-text author contribution statement towards a more formal statement based on Contributor Role Taxonomy (CRediT) terms, please remove the present Author Contribution section and instead specify each author's contribution(s) directly in the Author Information page of our submission system during upload of the final manuscript. See <https://casrai.org/credit/> for more information.
- Missing callout for Fig. 6E; callout for Table 2 should be renamed as Table 2 is not uploaded
- Please make sure that the author checklist is filled out (available for download on the manuscript page)

- Please provide figures in separate files. Main figures should be uploaded as individual Figure files, instead of being in a PDF, and up to 5 EV figures should be uploaded as individual, high-resolution Figure files and their figure legends should be included in ms file

- APPENDIX 1 FILE WITH ToC: Appendix file needs to be in PDF format; extra EV figures and Tables EV1 and EV2 should be compiled in Appendix PDF with the nomenclature Appendix Figure S# and Appendix Table S# with their legends placed below figures and above tables; title page should contain subtitle "Appendix for Inhibitors of eIF1A-ribosome interaction unveil uORF regulation and antitumor and antiviral effects" and ToC with page numbers

- Please provide the Reagent and Tools Table. For more information, please check <https://www.embopress.org/page/journal/14602075/authorguide#structuredmethods> and download the template for Reagent Table (attached for your convenience)

- Please provide suggestions for a short 'blurb' text prefacing and summing up the conceptual aspect of the study in two sentences (max. 250 characters), followed by 3-5 one-sentence 'bullet points' with brief factual statements of key results of the paper; they will form the basis of an editor-written 'Synopsis' accompanying the online version of the article. Please also provide an altered synopsis image, making sure that the aspect ratio conforms to our website's format - it should be exactly 550 pixels wide and between 300-600 pixels high.

- Author email bounced:

-- Daniel Hayat - daniel.hayat@wiedzmann.ac.il

-- Ariel Ogran - Ariel.Ogran@wiedzmann.ac.il

With best regards,

Cornelius Schneider

Cornelius Schneider, PhD
Editor
The EMBO Journal
c.schneider@embojournal.org

We generally allow three months as standard revision time. As a matter of policy, competing manuscripts published during this period will not negatively impact on our assessment of the conceptual advance presented by your study. However, we request that you contact the editor as soon as possible upon publication of any related work, to discuss how to proceed.

We realize that it is difficult to revise to a specific deadline. In the interest of protecting the conceptual advance provided by the work, we recommend a revision within 3 months (13th Apr 2025). Please discuss the revision progress ahead of this time with the editor if you require more time to complete the revisions. Use the link below to submit your revision:

Referee #1:

The authors have satisfactorily addressed most of my comments. Thank you. A few remaining minor comments are listed below.

1) Fig. 1C. A detailed map showing all uORFs of IRF7 would be helpful.

2) Fig. EV1A shows GUG but accord. to the text it should show CUG.

3) Fig. 4L - M. The 46 nt constructs +/- UAA actually tell us what is the contribution of leaky scanning vs. reinitiation to the overall activity measured. The contribution of leaky scanning to activities of both 46 and 95 nt constructs should be (is generally considered to be) the same. This is not apparent from the text; in fact the text is somewhat misleading in this aspect. Similarly, the text (highlighted in yellow) describing the new Figure 4N is somewhat awkwardly worded and should be clarified.

4) Previous point 5. I still find the chapter on SARS as the weakest point of the manuscript (the new data did not really convince me) and suggest at least to tone down the conclusions (e.g. removing them for the abstract etc.), as they do not add much to the whole story. Quite the contrary, they evoke a feeling of a nowadays fancy/trendy addition. However, the final decision is indeed on the authors.

5) Previous point 11. Despite the author's claim, this point was not addressed in the revised manuscript - to the best of my knowledge.

Referee #2:

The authors addressed all my concerns except the two following points:

My concerns were the following

- In Figure 2C, the presence of the introduced mRNA should also be monitored to ensure that the authors are looking at programmed-48S complex with their mRNA rather than endogenous 48S particles.

The authors responded to this comment that without RNA, the nice peak of 48S is not visible. This critical control should be included in this figure, namely the same experiment performed without added mRNA and the corresponding western blots against eIF1A, RPS10e and RPL36A should also be shown. The fact that the authors are using this approach since more than a decade does not free them from including genuine negative controls.

- In Figure 5B, the authors test the impact of the inhibitors on cell viability of OVCAR8 cells. A normal cell line should be included in this assay to ensure non-toxicity of the compounds to non-cancer cells.

The authors responded that the tested compounds affect proliferation of cancerous (OVCAR8) as well as noncancerous cells (MEFs and HEK293). This important issue should be mentioned and discussed in the manuscript.

Referee #4:

This manuscript is very interesting and unveils new mechanisms in translation initiation centered on eIF1A activity. There are several novel aspects of this work, especially the discovery of inhibitors for eIF1A-RPS10 interaction. The authors performed many mechanistic and functional experiments employing these inhibitors, leading to important conclusions on how eIF1A-RPS10 affects the translation initiation of specific mRNAs with unique start codon topology. The authors also performed experiments that will open new directions in the field to further explore the potential therapeutic benefit of inhibiting eIF1A-RPS10 interaction for cancer and viral infection treatment.

This reviewer appreciates that the authors addressed all the original reviewer's comments and added much more data to satisfactorily respond to the comments from Reviewer 3. I hope the authors can modify the discussion by reducing the repetition of the results. Instead, they should highlight their results in the growing appreciation of how translation specificity can be exerted by canonical translation factors and the importance of this translation specificity in the context of cancer and infection disease fields.

This manuscript is ready to be published in the Embo Journal.

Response to editorial queries

Figure Legends (main + EV):

1. Please note that the figure titles for figures EV 1; EV 3; EV 7.

Figure titles corrected.

2. Please note that the figure 2d is not provided in the manuscript, however the legend for the same is provided.

Corrected.

3. Please define the annotated p values **/* as well as provide the exact p-values for the same in the legend of figure 5d; as appropriate.

We checked and corrected the definitions of all p-values.

4. Please note that the exact p values are not provided in the legends of figures 1c-d, f; 3c; 4b, g, i, m-n; 6a, f; EV 1a-b; EV 7a; EV 8b-c.

We checked and corrected the definitions of all p-values.

5. Please indicate the statistical test used for data analysis in the legends of figures 5d; EV 7a.

WE checked and included statistical test as requested.

6. Please note that for the figure 6b, p-values are indicated in the legend. However, comparison for the same, ""**/*"" has not been represented in the figure. Also, provide exact p-value and statistical test for the same.

The asterisks were removed since the graph presents two biological replicates.

8. Please note that the scale bar is missing for figures 3a-b.

Scale bar was added.

9. Please note that axis gaps are not labeled appropriately in figure EV 8b.

Corrected.

10. There are only n=2 replicates in figures 4b; 6b and d. Please either add additional replicates or remove the statistical analysis from the figure, legends and text and display the individual datapoints in the figure.

As in Figure 4b, the control has 3 replicates, so it is possible to perform a statistical One-sample t-test, which is now defined in the legend. As for Figure 6b and d, the statistical tests were removed.

11. Although 'n' is provided, please describe the nature of entity for 'n' in the legends of figures 1c-d, f; 3c; 4b, i, m; 5b; 6a-b, f, h; EV 1a-d; EV 7a-b; EV 8b-c.

Authors' response: "As for your point #4, please note that for each of these sections, the nature of the entity is already described in the legends."

Editor: Please specify if the indicated n corresponds to independent biological replicates.

We added the definition of “n” as the number of independent biological replicates to all figure legends.

12. Please note that the measure of center for the error bars needs to be defined in the legends of figures 4b; 6a-b.

The information is now included in legends.

Additional editorial notes:

- On the abstract page of the manuscript, please include 4-5 general keyword terms to enhance searchability.

Keywords were added on the abstract page.

- Please rename the Conflict of Interest section into "Disclosure and Competing Interests Statement", in accordance with our updated Guide to Authors

(<https://www.embopress.org/competing-interests>)

The Conflict of Interest section was renamed into "Disclosure and Competing Interests Statement"

- As we are switching from a free-text author contribution statement towards a more formal statement based on Contributor Role Taxonomy (CRediT) terms, please remove the present Author Contribution section and instead specify each author's contribution(s) directly in the Author Information page of our submission system during upload of the final manuscript. See <https://casrai.org/credit/> for more information.

Author contribution statement was removed from the main text.

- Missing callout for Fig. 6E; callout for Table 2 should be renamed as Table 2 is not uploaded

Corrected.

- Please make sure that the author checklist is filled out (available for download on the manuscript page)

Done

- Please provide figures in separate files. Main figures should be uploaded as individual Figure files, instead of being in a PDF, and up to 5 EV figures should be uploaded as individual, high-resolution Figure files and their figure legends should be included in ms file

Done

- APPENDIX 1 FILE WITH ToC: Appendix file needs to be in PDF format; extra EV figures and Tables EV1 and EV2 should be compiled in Appendix PDF with the nomenclature Appendix Figure S# and Appendix Table S# with their legends placed below figures and above tables; title page should contain subtitle "Appendix for Inhibitors of eIF1A-ribosome interaction unveil uORF regulation and antitumor and antiviral effects" and ToC with page numbers

Done

- Please provide the Reagent and Tools Table. For more information, please check <https://www.embopress.org/page/journal/14602075/authorguide#structuredmethods> and download the template for Reagent Table (attached for your convenience)
The reagent and Tools Table was downloaded and filled.

- Please provide suggestions for a short 'blurb' text prefacing and summing up the conceptual aspect of the study in two sentences (max. 250 characters), followed by 3-5 one-sentence 'bullet points' with brief factual statements of key results of the paper; they will form the basis of an editor-written 'Synopsis' accompanying the online version of the article. Please also provide an altered synopsis image, making sure that the aspect ratio conforms to our website's format - it should be exactly 550 pixels wide and between 300-600 pixels high.

We provide the synopsis and image as per the guidelines.

- Author email bounced:

-- Daniel Hayat - daniel.hayat@wizmann.ac.il

-- Ariel Ogran - Ariel.Ogran@wizmann.ac.il

The emails should be corrected to the following

Daniel Hayat - daniel.hayat@weizmann.ac.il

Ariel Ogran - Ariel.Ogran@weizmann.ac.il

Referee #1:

The authors have satisfactorily addressed most of my comments. Thank you. A few remaining minor comments are listed below.

1) Fig. 1C. A detailed map showing all uORFs of IRF7 would be helpful.

A scheme for IRF7 5'UTR was added (Figure 1C).

2) Fig. EV1A shows GUG but accord. to the text it should show CUG.

Thanks for pointing out this error, corrected to CUG.

3) Fig. 4L - M. The 46 nt constructs +/- UAA actually tell us what is the contribution of leaky scanning vs. reinitiation to the overall activity measured. The contribution of leaky scanning to activities of both 46 and 95 nt constructs should be (is generally considered to be) the same. This is not apparent from the text; in fact the text is somewhat misleading in this aspect. Similarly, the text (highlighted in yellow) describing the new Figure 4N is somewhat awkwardly worded and should be clarified.

We thank the referee for this constructive comment. The text was revised for better clarity.

4) Previous point 5. I still find the chapter on SARS as the weakest point of the manuscript (the new data did not really convince me) and suggest at least to tone down the conclusions (e.g. removing them for the abstract etc.), as they do not add much to the whole story. Quite the contrary, they evoke a feeling of a nowadays fancy/trendy addition. However, the final decision is indeed on the authors.

We appreciate this suggestion but disagree with respect to the importance of this experiment. The SARS-CoV-2 experiments present a specific example of a translation regulatory feature regulated by RPS10, which could only be discovered through the use of Rps10-selective inhibitors.

5) Previous point 11. Despite the author's claim, this point was not addressed in the revised manuscript - to the best of my knowledge.

We now further expanded the discussion on the points raised in the previous point 11 specifically with respect to the possibility that eIF1A can bind with lower affinity to other components of the PIC in the presence of 1Ais and the possible mechanism of actin of the RPS10-specific 1Ais.

Referee #2:

The authors addressed all my concerns except the two following points:

My concerns were the following

- In Figure 2C, the presence of the introduced mRNA should also be monitored to ensure that the authors are looking at programmed-48S complex with their mRNA rather than endogenous 48S particles.

The authors responded to this comment that without RNA, the nice peak of 48S is not visible. This critical control should be included in this figure, namely the same experiment performed without added mRNA and the corresponding western blots against eIF1A, RPS10e and RPL36A should also be shown. The fact that the authors are using this approach since more than a decade does not free them from including genuine negative controls.

We have now added the suggested control experiment in which RRL was incubated with and without mRNA, demonstrating the appearance of the 48S peak only in the presence of mRNA (Fig. EV 4A). The fractions were also subjected to RNA analysis and western blots to confirm appropriate 48S formation (Fig. EV 4A).

- In Figure 5B, the authors test the impact of the inhibitors on cell viability of OVCAR8 cells. A normal cell line should be included in this assay to ensure non-toxicity of the compounds to non-cancer cells.

The authors responded that the tested compounds affect proliferation of cancerous (OVCAR8) as well as noncancerous cells (MEFs and HEK293). This important issue should be mentioned and discussed in the manuscript.

Thank you, this is now added to the text (p. 10, bottom).

Dear Prof. Dikstein,

I am pleased to inform you that your manuscript has been accepted for publication in the EMBO Journal.

Yours sincerely,

Cornelius Schneider, PhD
Editor
The EMBO Journal
c.schneider@embojournal.org
